

# Ozone anomalies over polar regions during the stratospheric warming events

Guochun Shi[1,2], Witali Krochin[1,2], Eric Sauvageat[3], and Gunter Stober[1,2]

[1]Oeschger Center for Climate Change Research, University of Bern, Bern, Switzerland
[2]Institute of Applied Physics, University of Bern, Bern, Switzerland
[3]Federal Office of Meteorology and Climatology, MeteoSwiss, Payerne, Switzerland

**Correspondence:** Guochun Shi (guochun.shi@unibe.ch)

**Abstract.** The impact of major sudden stratospheric warming (SSW) events and early final stratospheric warming (FSW) events on ozone variations in the middle atmosphere in the Arctic is investigated by performing microwave radiometer measurements above Ny-Ålesund, Svalbard (79° N, 12° E) with GROMOS-C. The retrieved daily ozone profiles during SSW and FSW events in the stratosphere and lower mesosphere at 20-70 km from microwave observations are cross-compared to MERRA-2 and MLS. The vertically resolved structure of polar ozone anomalies relative to the climatologies derived from GROMOS-C, MERRA-2, and MLS shed light on the consistent pattern in the evolution of ozone anomalies during both types of events. For SSW events, ozone anomalies are positive throughout all altitudes within 30 days after the onset, followed by negative anomalies descending downward in the middle stratosphere. However, positive anomalies in the middle and lower stratosphere and negative in the upper stratosphere at onset are followed by negative anomalies with descending in the middle stratosphere and positive anomalies in the upper stratosphere during FSW events. We document the underlying dynamical and chemical mechanisms that are responsible for the observed ozone anomalies in the entire life cycle of SSW and FSW events. Polar ozone anomalies in the lower and middle stratosphere undergo a rapid and long-lasting increase of more than 1 ppmv close to SSW onset, which is attributed to the dynamical processes of the horizontal eddy effect and vertical advection. This response pattern for FSW events is associated with the combined effects of dynamical and chemical terms, which reflect the photochemical processes counteracted partially by positive horizontal eddy transport, in particular in the middle stratosphere. Here, we contrast results from MERRA-2 reanalysis and chemistry-climate models to quantify the impact of dynamical and chemical processes on ozone anomalies during SSW and FSW events. In addition, we find that the variability in polar total column ozone (TCO) is associated with horizontal eddy transport and vertical advection of ozone in the lower stratosphere. This study enhances our understanding of the mechanisms that control changes in polar ozone during the life cycle of SSW and FSW events, providing a new aspect to quantitative analysis of dynamical and chemical fields.





# 1  Introduction

The wintertime polar stratosphere is characterized by a strong, westerly, and cold polar vortex. Due to the different sea-land distribution in the northern and southern hemispheres large-scale waves with several hundred kilometers of wavelength are

generated in the troposphere. These waves propagate upward into the stratosphere, disturbing or weakening the polar vortex, thus, affecting the dynamics there (Andrews et al., 1987). The occurrence of SSW event (Charlton and Polvani, 2007; Butler et al., 2015) during mid-winter is mainly attributed to the split or displacement of the stratospheric polar vortex by the upward-propagating planetary waves (Holton, 1980; Pancheva et al., 2008; Matthias et al., 2013; Albers and Birner, 2014; Qin et al., 2021; Baldwin et al., 2021). Observed FSW event (Black and McDaniel, 2007) in early spring depends on variations in the

upward propagation of tropospheric planetary waves, as well as increasing shortwave radiation in the polar region (Salby and Callaghan, 2007; Sun et al., 2011; Thiéblemont et al., 2019). As the only atmosphere species effectively absorbing ultraviolet solar radiation from about 250-300 nm, ozone plays the most important role in the coupling between chemistry, radiation, and dynamical processes. Therefore, the dynamical fluctuations and chemical reactions of stratospheric ozone in the Arctic are subject to both events (Lubis et al., 2017; Oehrlein et al., 2020; Friedel et al., 2022).

SSW events characterized by abrupt warming and weakening or reversal of the polar wintertime westerly circulation lead to extreme ozone variability at the polar latitudes (Schranz et al., 2019, 2020). de la Cámara et al. (2018) utilized the Whole Atmosphere Community Climate Model (WACCM) output and European Centre for Medium-Range Weather Forecasts Re-Analysis Interim (ERAI) to display a comprehensive and quantitative analysis of ozone advective transport and mixing in equivalent latitude coordinates during the life cycle of SSW with Polar-night Jet Oscillation and without Polar-night Jet Os-

cillation. Bahramvash Shams et al. (2022) emphasized the high variability of middle stratosphere ozone fluctuations and the key role of vertical advection in mid-stratospheric ozone during SSW using MERRA-2 reanalysis data. Oehrlein et al. (2020) analyzed the response with and without interactive chemistry versions of WACCM to major SSW events, which resulted in a pattern resembling a more negative North Atlantic Oscillation following mid-winter SSW events. Hong and Reichler (2021) examined the changes of ozone in both the Arctic and the tropic regions and documented the underlying dynamical mecha-

nisms for the observed changes during the life cycle of SSW and vortex intensification events.

Several case studies of FSW events, utilizing a combination of chemistry-climate models and reanalysis data emphasize stratospheric ozone anomalies, which are influenced by the position and strength of the polar vortex and chemical processing to different dynamical conditions. Salby and Callaghan (2007) used a three-dimensional model of dynamics and photochemistry to investigate the enriched polar ozone during springtime through isentropic mixing by planetary waves and eliminated much

of the apparent ozone depletion. Thiéblemont et al. (2019) confirmed the timing of FSW affected by the ozone and greenhouse gases via coupled chemistry-climate models of WACCM. Lawrence et al. (2020) used MERRA-2 and the Japanese Meteorological Agency's 55-year reanalysis (JRA-55) to display ozone depletion and TCO amounts in the northern hemisphere polar cap decreasing to the lowest ever observed in springtime. Hong and Reichler (2021) investigates the persistent loss in Arctic ozone during vortex intensifications, which is dramatically compensated by sudden warming-like increases after the final

warming. Friedel et al. (2022) contrasted results from chemistry-climate models with and without interactive ozone chemistry



to quantify the impact of ozone anomalies on the timing of the FSW and its effects on surface climate. Other studies focussed mainly on the dynamical effects of the FSW leveraging ground-based and satellite observations to characterize the transition to the spring and summer circulation concerning the dynamical or radiative forcing (Matthias et al., 2021) or the tidal amplification of the semidiurnal tide in the aftermath of major SSW evens (Stober et al., 2020).

Utilizing the outcomes of ozone continuity equations, we derive the relative contributions of dynamical transport versus chemical processes in determining the polar ozone anomalies behavior observed in SSW and FSW events. In addition, we disclose that polar ozone anomalies in the lower stratosphere mainly predominantly governed by dynamical processes exhibit a strong correlation with polar total column ozone. Overall, our goal is not only to provide a new view of the dynamical and chemical-driven variability in polar ozone anomalies but also to apply it to the validation of coupled chemistry–climate models and other

reanalysis data.

The paper is structured as follows: Section 2 describes the data and methods. Section 3 provides the vertically resolved ozone field at polar latitude stations. Section 4 discusses the ozone budget through dynamical and chemical processes and the dynamically controlled polar TCO. Finally, the results are summarized and discussed in Section 5.

## 2  Data and methods

### 70  2.1  GROMOS-C

GROMOS-C (GRound-based Ozone MOnitoring System for Campaigns) is an ozone microwave radiometer that measures the ozone emission line at 110.836 GHz at Ny-Ålesund, Svalbard (79° N, 12° E) since September 2015. It was built by the Institute of Applied Physics at the University of Bern. The radiometer is very compact and optimized for autonomous operation. Hence, it can be transported and operated at remote field sites under extreme climate conditions. GROMOS-C observes subsequently

on the four cardinal directions (north, east, south, and west) under an elevation angle of 22° with a sampling time of 4 s. Ozone VMR profiles are retrieved from the ozone spectra with a temporal averaging of 2 hours leveraging the Atmospheric Radiative Transfer Simulator version-2 (ARTS2; Eriksson et al., 2011) and Qpack2 software (Eriksson et al., 2005) according to the optimal estimation algorithm (Rodgers, 2000). An apriori ozone profile is required for optimal estimation and is taken from an MLS climatology collected between the years 2004-2013. The retrieved 2 hourly ozone profiles have a vertical resolution

of 10-12 km in the stratosphere and up to 20 km in the mesosphere, which cover a sensitive altitude range of 23-70 km. The averaging kernels (AVKs) of GROMOS-C together with its measurement response, errors, and ozone profiles are shown in Fernández et al. (2015).

### 2.2  Aura-MLS

NASA's Earth Observing System (EOS) Microwave Limb Sounder (MLS) instruments on board the Aura spacecraft measure

thermal emissions from the limb of Earth's atmosphere. MLS provides comprehensive measurements of vertical profiles of temperature and 15 chemical species from the upper troposphere to the mesosphere, spanning nearly pole-to-pole coverage



from 82°S to 82°N (Waters et al., 2006).

The ozone profile is retrieved using the 240 GHz microwave band, which extends from 261 hPa to 0.0215 hPa on 38 pressure layers. Vertical spacing for these layers is about 1.3 km everywhere below 1 hPa and about 2.7 km at most altitudes above 1 hPa. The vertical resolution of the retrieved ozone profile is reported to be around 3 km extending from 261 hPa up into the mesosphere (Livesey et al., (last access: 17 December 2023; Schwartz et al., 2015a). The time records for the MLS ozone profiles used in this study are from August 2004 to December 2021 (in the subsequent section, the SSW event in 2003/2004 is not applicable for analyzing ozone variations from MLS). MLS passes at Ny-Ålesund twice a day at around 04:00 and 10:00 UTC. Profiles for comparison are extracted if the location is within ±1.2° latitude and ±6° longitude of Ny-Ålesund.

## 2.3 MERRA-2

The Modern-Era Retrospective Analysis for Research and Applications, version 2 (Waters et al., 2006; Gelaro et al., 2017, MERRA-2) of the Goddard Earth Observing System-5 (GEOS-5) atmospheric general circulation model (AGCM) is the latest global atmospheric reanalysis produced by the NASA Global Modeling and Assimilation Office (GMAO) from 1980 until present. A variety of data sets are assimilated into AGCM to create 3-dimensional MERRA-2 ozone datasets with a time resolution of 3 hours (Wargan et al., 2017; Gelaro et al., 2017). The retrieved ozone profiles from the Solar Backscatter Ultraviolet Radiometer (SBUV, 1980 to 2004) and the MLS (since August 2004, down to 177 hPa until 2015, down to 215 hPa after 2015) and TCO from SBUV (1980 to 2004) and the Ozone Monitoring Instrument (OMI) (since 2004) are used to estimate ozone in MERRA-2 (Gelaro et al., 2017).

MERRA-2 data has been used to study ozone trends, processes, and validations with ozonesondes, microwave radiometers, and satellite observations (Lubis et al., 2017; Albers et al., 2018; Wargan et al., 2018; Schranz et al., 2020; Hong and Reichler, 2021; Bahramvash Shams et al., 2022; Shi et al., 2023). In this study, the ozone dataset from MERRA-2 reanalysis with 72 model levels from the surface to 0.01 hPa and a horizontal resolution of $0.5° \times 0.625°$ and a time resolution of 3 hours will be used. MERRA-2 ozone at the model levels complements our microwave measurements to investigate the polar ozone variations in the stratosphere and mesosphere. Meteorological variables such as temperature, zonal and meridional wind, and vertical pressure velocity extracted from 42 pressure levels facilitate the calculation of variables such as residual circulation. Given by Lubis et al. (2017), the MERRA-2 ozone tendency product on 42 pressure levels from the surface up to 0.1 hPa (https://doi.org/10.5067/S0LYTK57786Z) is modeled with the GEOS-Chemistry transport model by using odd-oxygen mixing ratio, $qO_x$, as its prognostic variable (Bosilovich, 2015). The model is driven by assimilated meteorological data from the GEOS of the GMAO (Stajner et al., 2008; Wargan et al., 2015). This includes an odd-oxygen family transport model that provides the ozone concentration necessary for solar absorption. The vertically integrated ozone tendency is given as (Bosilovich, 2015; Wargan et al., 2017):

$$[\frac{\partial \overline{qO_x}}{\partial t}]_{TOT} = [-\nabla \cdot (\overline{vqO_x})]_{DYN} + [\frac{\partial \overline{qO_x}}{\partial t}]_{PHY} + [\frac{\partial \overline{qO_x}}{\partial t}]_{ANA} \tag{1}$$





The dynamical contribution to the total ozone tendency (TOT) is the convergence of odd-oxygen mixing ratio products (the
first right-hand-side term (DYN) of Eq. (1)). The total physics product (PHY) includes the parameterized production and loss
terms, mostly in the stratosphere, and the analysis product (ANA) is the corrected ozone tendency from data analysis. The
analysis term is part of the incremental analysis update, which is used in the GEOS-5 model and is an additional constraint of
the model due to observations. The total ozone tendency from physics (PHY) is decomposed into contributions from chemistry
(CHM), turbulence (TRB), and moist physics (MST). Given a parameterized ozone chemistry in MERRA-2, the total ozone
tendency from chemistry (CHM) is analyzed together with the correcting tendency term (i.e., CHM+ANA). The contributions
of turbulence and moist physics are negligible in the stratosphere and are therefore not considered in this analysis.

## 2.4   TEM ozone budget

The local changes for atmospheric tracers ($\overline{\chi}$) are investigated using the transformed Eulerian mean (TEM) continuity equation
that results from transport processes and chemical sources and sinks as follows (Andrews et al., 1987):

$$\overline{\chi}_t = \underbrace{-\overline{v}^*\overline{\chi}_y - \overline{\omega}^*\overline{\chi}_z + e^{z/H}\nabla \cdot M}_{\overline{\chi}_{dyn}} + \underbrace{(\overline{P} - \overline{L})}_{\overline{S}} \tag{2}$$

where $\overline{\chi}_t$ is the tracer tendency that denotes transport of the zonal mean tracers volume mixing ratios due to the horizontal
and vertical advection by the residual circulation ($\overline{v}^*$, $\overline{\omega}^*$), the horizontal and vertical eddy transport effects ($e^{z/H}\nabla \cdot M_y$,
$e^{z/H}\nabla \cdot M_z$), and $\overline{S}$ is chemical production minus loss ($P - L$). The chemical net is calculated as the residual of the left side
minus the sum of the first four terms $\overline{\chi}_{dyn}$ on the right side of Eq. (2) to better understand the chemical component in the
stratosphere. The overbars indicate zonal means and primes denote the departure from the zonal mean. The scale height is
represented by $H$ of 7 km.

The $\overline{v}^*$ and $\overline{\omega}^*$ in Eq. (2) denote the TEM residual meridional and vertical winds defined as:

$$\overline{v}^* = (\overline{v} - e^{z/H} \partial_z(e^{z/H}\overline{v'\theta'}/\overline{\theta}_z) \tag{3}$$


$$\overline{\omega}^* = \overline{\omega} + (a\cos\varphi)^{-1} \partial_\varphi(\cos(\varphi)\overline{v'\theta'}/\overline{\theta}_z) \tag{4}$$

where $v$ and $\omega$ are the meridional and vertical winds, $\theta$ is the potential temperature, $a$ is the earth's radius, and $\varphi$ is the latitude.
Here, the eddy transport vector $M$ can be decomposed into meridional and vertical components $M_y$ and $M_z$ respectively
(Andrews et al., 1987):

$$M_y = -e^{z/H}(\overline{v'\chi'} - \overline{v'\theta'}/\overline{\theta}_z\overline{\chi}_z) \tag{5}$$

$$M_z = -e^{z/H}(\overline{\omega'\chi'} - \overline{v'\theta'}/\overline{\theta}_z\overline{\chi}_y) \tag{6}$$



## 2.5 Identification of SSW and FSW events

Stratospheric warming events are a crucial stratospheric phenomenon and indicate the vertical coupling of the entire middle atmosphere affecting the mesosphere, stratosphere, and troposphere. Many studies combined temperature increases and wind reversals to detect major SSW events in midwinter (Charlton and Polvani, 2007; Hu et al., 2014; Butler et al., 2015; Butler and Gerber, 2018). One of the most often-used definitions of a major SSW during wintertime (Charlton and Polvani, 2007) is that the zonal-mean zonal winds at 60°N and the 10 hPa level reverses from westerly to easterly and the zonal-mean temperature gradient between 60°N and 90°N becomes positive. As shown in Table 1, we identify 10 major SSW events in this study as described in Li et al. (2023). For the FSW events, different studies have analyzed springtime stratospheric zonal winds by single pressure levels at varying latitudes and thresholds (Black and McDaniel, 2007; Byrne et al., 2017; Matthias et al., 2021) and multiple pressure levels (Hardiman et al., 2011). We found 7 early FSW events (in Table 1) identified by Butler and Domeisen (2021) based on the criterion that the daily mean zonal-mean zonal winds at 60°N latitude and 10 hPa exhibit an easterly flow and remain so continuously for more than 10 consecutive days, as outlined by Butler and Gerber (2018). The SSW and FSW composites will be discussed both for the wind and temperature fields and anomalies which are defined as deviations from the daily seasonal climatology.

**Table 1.** Dates of major SSW and early FSW events were used for the composite in this study.

| Number | Winters | SSW central date | Winters | FSW central date |
| --- | --- | --- | --- | --- |
| 1 | 2003/2004 | 05 Jan 2004 | 2004/2005 | 13 Mar 2005 |
| 2 | 2005/2006 | 21 Jan 2006 | 2010/2011 | 04 Apr 2011 |
| 3 | 2006/2007 | 24 Feb 2007 | 2013/2014 | 27 Mar 2014 |
| 4 | 2007/2008 | 22 Feb 2008 | 2014/2015 | 28 Mar 2015 |
| 5 | 2008/2009 | 24 Jan 2009 | 2015/2016 | 05 Mar 2016 |
| 6 | 2009/2010 | 09 Feb 2010 | 2016/2017 | 08 Apr 2017 |
| 7 | 2012/2013 | 06 Jan 2013 | 2019/2020 | 14 Mar 2020 |
| 8 | 2017/2018 | 12 Feb 2018 | - | - |
| 9 | 2018/2019 | 02 Jan 2019 | - | - |
| 10 | 2020/2021 | 03 Jan 2021 | - | - |



# 3 Meteorological background situations

To examine the ozone anomalies during late winter over the polar latitude station, we summarize some key dynamical quantities of SSW and FSW events. Fig. 1 illustrates the pressure-time evolution of the SSW and FSW composite zonal-mean zonal wind (at 60°N) and temperature (70° - 90° N) in MERRA-2 reanalysis data. Below approximately 0.1 hPa, the westerly wind rapidly

weakens lags - 10 days and then switches to an easterly wind after the SSW onset (lags 0 days) in Fig. 1a. The easterly wind returns after approximately 15 days and strengthens to a maximum speed of 80 m s$^{-1}$ around lags 35 days. The temperature fields undergo alterations in conjunction with the wind field reversal. The SSW onset is characterized by the rapid warming in the stratosphere in Fig. 1c, indicating the rapid descent of the stratopause to lower altitudes. In Fig. 1b, the zonal-mean zonal wind at 60°N and 10 hPa is easterly with lags 50 days until the early summer and does not reverse to westerly. The wind

reversal is accompanied by a temperature increase exceeding 280 K after the FSW onset in Fig. 1d.

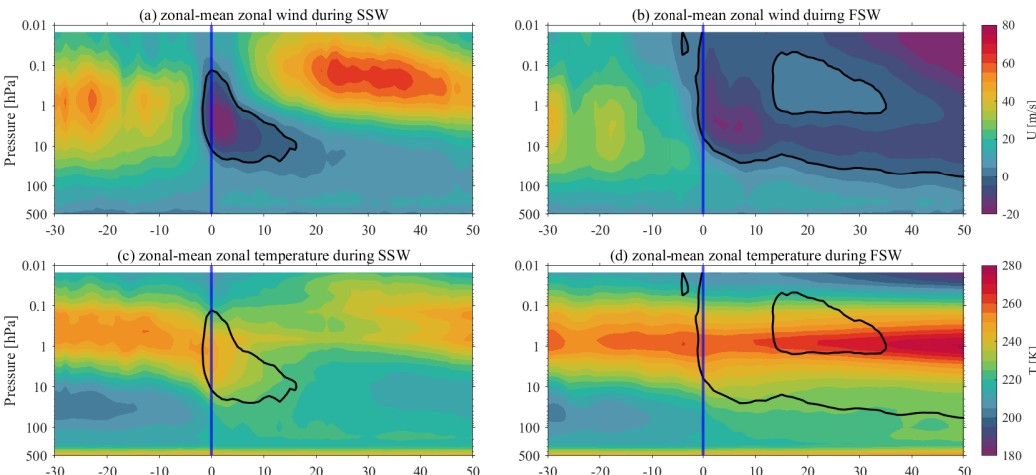

**Figure 1.** Pressure-time section of SSW and FSW composite zonal-mean (a, b) zonal wind, (c, d) temperature from MERRA-2. Time is relative to the SSW and FSW onset on the abscissa. The vertical blue line represents the onset day (day 0). The zero wind contour is in black.

Furthermore, we derive anomalies for the relevant physical quantities by subtracting the mean climatology obtained from all years for our composites of SSW and FSW events. Significant anomalies of wind and temperature and $\overline{\omega}^*$ extend nearly over the entire pressure range and throughout the life cycle of SSW and FSW in Fig. 2. The strongest negative wind anomalies occur

during the first 15 days after the SSW onset and diminish within 20 days in the stratosphere corresponding to the strongest positive anomalies in the mesosphere occurring between 20 to 50 days after the onset day as shown in Fig. 2a. These changes occur in parallel to rapid stratospheric warming, with the temperature maxima appearing in near vertical quadrature with the wind anomalies during those 5 days before and after the SSW onset (in Fig. 2c). During the recovery phase following the SSW, progressively descending negative anomalies in the stratosphere appear with positive anomalies in the mesosphere, along with

the reformation of the 'normal' stratopause. The lower mesosphere exhibits negative wind anomalies ranging from lags of -30





to 20 days, with the most pronounced negative values observed at 1 hPa (Fig. 1a). The vertical extent of the zonal wind and temperature anomalies at FSW onset is similar to the SSW event, but the magnitude and strength are different. The temperature anomaly at 1 hPa almost vanishes and remains around zero after FSW onset. $\overline{\omega}^*$ anomalies over the polar regions (70-90°N) as an indicator of wave forcing show more intense downward propagation (blue) and upward propagation (red) during both events. The obvious difference between the two types of events is that strong upwelling starts about three weeks earlier at negative lags for SSW events (Fig. 2e). The statistically significant positive anomalies vanish after 15 days, giving way to negative anomalies emerging within 30 days in Fig. 2e. In contrast, FSW events exhibit $\overline{\omega}^*$ anomalies to remain positive for a duration near 40 days above 1 hPa (Fig. 2f).

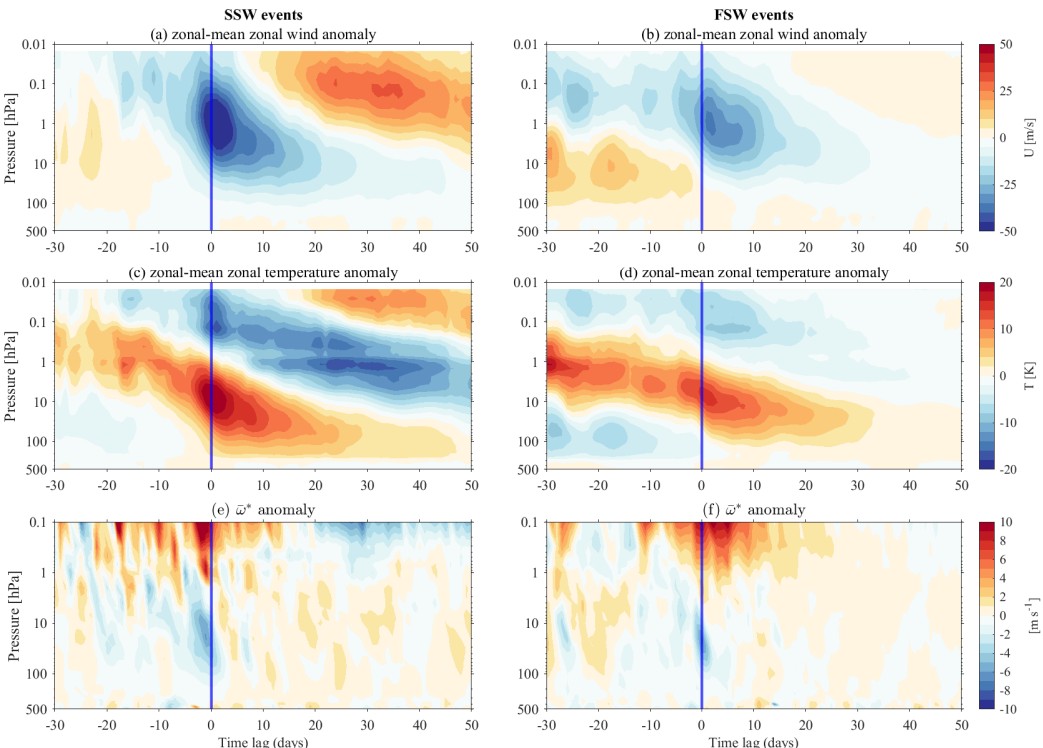

**Figure 2.** Pressure-time section of SSW and FSW composite zonal-mean (a, b) zonal wind anomalies, (c, d) temperature anomalies, (e, f) the vertical component of the residual circulation $\overline{\omega}^*$ anomalies for averaged polar regions (70° - 90° N) from MERRA-2.

## 4   Local changes over Ny-Ålesund, Svalbard (79° N, 12° E)

Leveraging continuous ozone measurements from the ground-based radiometer GROMOS-C at Ny-Ålesund, Svalbard (79° N, 12° E) and combining MERRA-2 and MLS datasets, we analyze the temporal evolution of ozone and provide more details on the impacts of SSWs. The main benefit of the ground-based observations is the much higher temporal resolution of two hours, which permits to estimate of the sampling bias from the satellite MLS taking data only at two local times. This higher temporal





resolution is also sufficient to resolve the daily ozone cycle (Schranz et al., 2018). Fig. 3 exhibits the SSW and FSW composite
ozone VMR at Ny-Ålesund (79° N, 12° E) as a function of time lag for the event central date. GROMOS-C measured ozone
VMR over Ny-Ålesund is greatly enhanced after an SSW and FSW onset. The results indicate a good agreement between
MERRA-2 (below 0.1 hPa) and MLS with GROMOS-C observations. However, due to the complexity of altered dynamics in
the winter polar regions introducing extra uncertainties into numerical models and data assimilation systems (Wargan et al.,
2017), ozone VMRs exhibit dramatic variability (in Fig. 3b, e) in the mesosphere from MERRA-2.

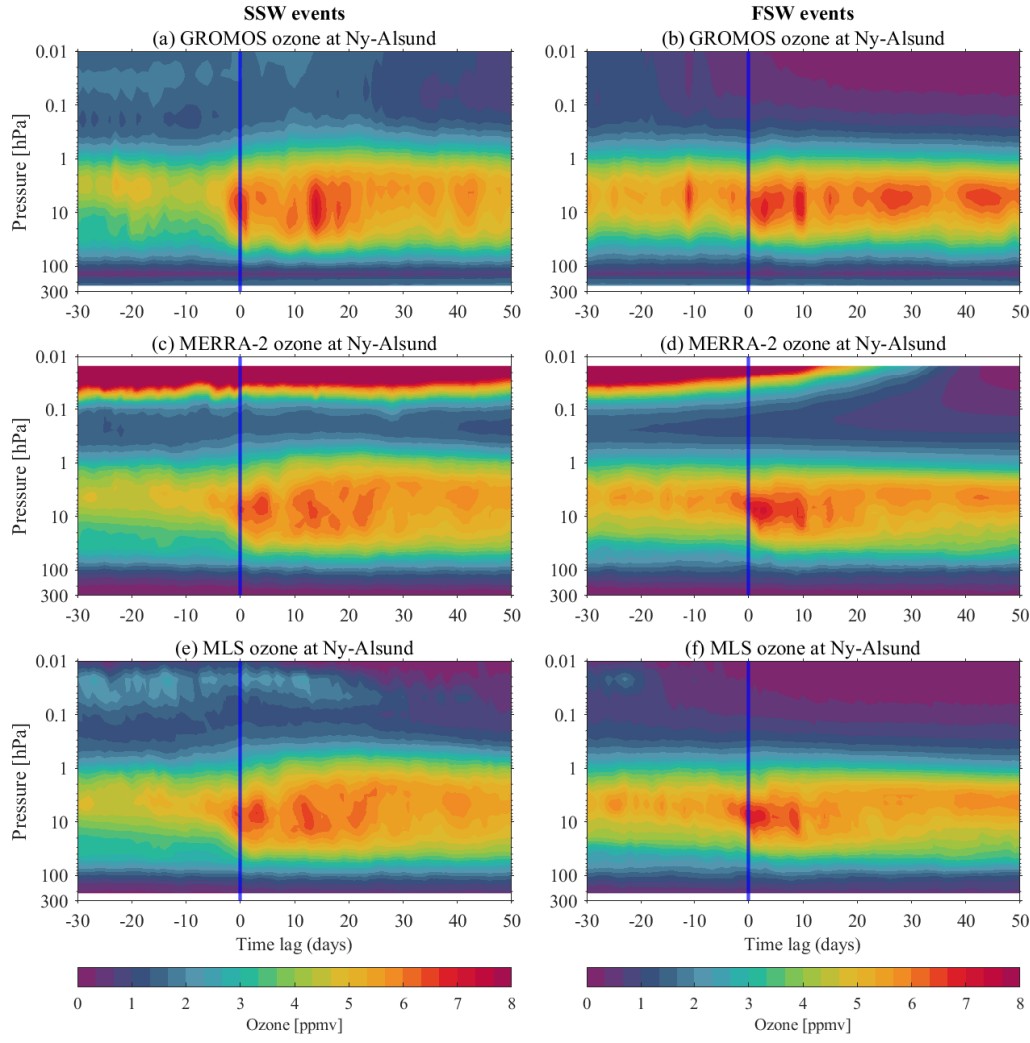

**Figure 3.** Pressure-time section of SSW and FSW composite ozone VMR from (a, d) GROMOS-C, (b, e) MERRA-2 and (c, f) MLS at
Ny-Ålesund (79° N, 12° E), respectively.



Fig. 4 shows the composite ozone anomalies in GROMOS-C, MERRA-2, and MLS at Ny-Ålesund (79° N, 12° E) as a function of time lag for the SSW and FSW central date. These subplots show very similar behavior despite the variety of data sets and years covered. The strongest positive ozone anomalies of up to 1 ppmv for more than 30 consecutive days after the SSW onset in the middle and upper stratospheric layers are evident. The positive ozone anomalies persist around 20 days after FSW onset in the middle stratosphere following a negative value of around 0.6 ppmv with descending downward in the upper

stratosphere. Otherwise, there is a negative ozone anomaly in the lower stratosphere and upper stratosphere before FSW onset which is stronger than SSW events. Note that the GROMOS-C composite is based on only three SSW and FSW events because the measurement campaign started in September 2015. The anomalies are estimated concerning the daily climatology (between 2015 and 2022).

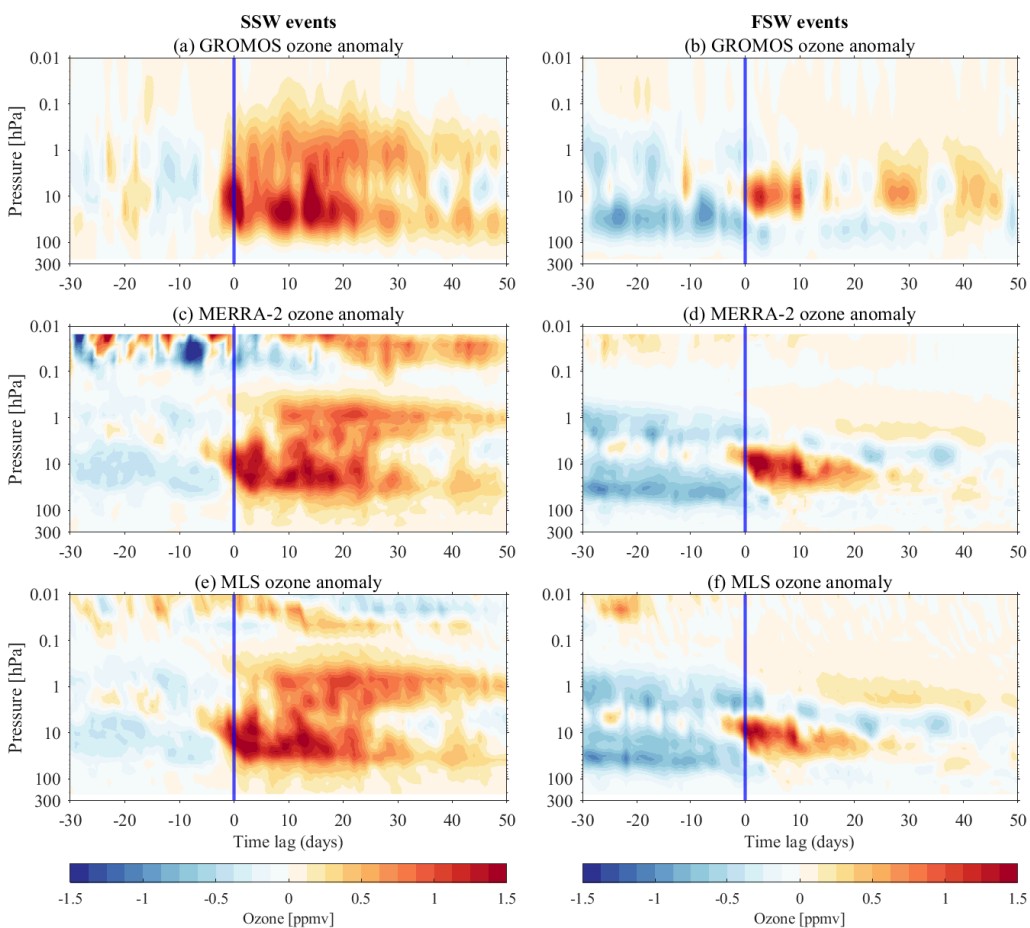

**Figure 4.** Same as Fig. 3, but for anomalies at Ny-Ålesund, Svalbard (79° N, 12° E)





## 5 Dynamical and Chemical effects on Ozone

### 5.1 Seasonal Cycle of Ozone Transport Budget

Seasonal changes in ozone from the eddy effect, advection transport, and chemical loss and production are shown in Fig. 5. The contribution from each term in Eq. (2) is calculated to infer the seasonal cycle of the ozone transport budget in MERRA-2 averaged between 70°N and 90°N. The ozone tendency $\overline{\chi}_t$ shows a distinct increase during the winter and fall and a decrease in the spring and early summer. In Fig. 5b, the horizontal eddy transport exhibits a prominent seasonal cycle, with positive and significantly stronger ozone eddy mixing observed throughout the stratosphere over the entire year, except for the summer months. Vertical eddy transport tends to show a dipole pattern with the opposite effects between the middle and upper stratosphere from the late fall to the early spring in Fig. 5c. This indicates that seasonality in eddy mixing plays a major role in the polar ozone annual cycle. Horizontal advection of ozone is much smaller compared to eddy mixing and chemical processes as shown in Fig. 5d (5-times increased). However, it has negative effects on the ozone mixing transport in the polar regions. During fall and winter, the vertical advection transport exhibits a comparable pattern to the vertical eddy term, yet it demonstrates a tendency towards positive values in the upper stratosphere during summer. Finally, the chemical term is positive in the wintertime in the middle stratosphere. However, the greatest ozone destruction occurs in spring reaching its maximum in April.

The lowest panels in Fig.5(g-l) show the seasonal cycle of all terms averaged over the latitudes from 70–90° N for the pressure levels of 10 hPa ($\sim 30\,\text{km}$), 3 hPa K ($\sim 40\,\text{km}$), and 1 hPa ($\sim 50\,\text{km}$). At 10 hPa horizontal eddy transport and net chemical loss nearly balance each other, particularly from February to Jun. Vertical eddy transport makes a negative contribution from September to April. Horizontal eddy transport has a large positive contribution within 0.4 ppmv day$^{-1}$ in March at 3 hPa, corresponding to maximum chemical ozone destruction. However, chemical production starts from October to February and has a peak in January at 3 hPa. Thus, the shape of the ozone seasonal cycle is mainly determined by the seasonally varying eddy mixing transport and chemical loss and production. At 1 hPa, the chemical term is of crucial relevance, the seasonal budget of ozone is completely controlled by competing effects of horizontal eddy transport and chemical term. As a result, the eddy mixing effectively transports ozone into the polar region during winter and spring, where the horizontal eddy transport is so large that it balances a large fraction of the chemical ozone destruction.



**Figure 5.** Seasonal cycle of the ozone tendencies as a function of time and pressure from MERRA-2 (period 2004-2021): (a) ozone tendency $\overline{\chi}_t$ and (b) horizontal eddy transport $-\overline{v}^*\overline{\chi}_y$, (c) vertical eddy transport $-\overline{\omega}^*\overline{\chi}_z$, (d) horizontal advection transport $e^{z/H}\nabla \cdot M_y$, (e) vertical advection transport $e^{z/H}\nabla \cdot M_z$, and (f) chemical loss and production $\overline{S}$ averaged the polar regions (70 - 90° N) based on Eq. (2). The third row is the comparison of each term at different pressure levels: (g) 10 hPa, (h) 3 hPa, and (i) 1 hPa.

Although the chemical term $\overline{S}$ displays the features of a chemical sink and source term, including location and seasonality in Fig. 5, there are differences compared to other methods of calculating ozone loss rates as shown in Fig. 6. We use the output from the chemistry transport model to display the seasonal cycle of total ozone tendency $TOT$, and due to dynamics $DYN$ and chemistry $CHM$ based on Eq. (1). $TOT$ shows good agreement in magnitude with the results $\overline{\chi}_t$ from the TEM analysis. The largest discrepancy between $\overline{S}$ and $CHM$ (between dynamical terms $DYN$ and chemical processes $CHM$)

occurs during the winter months in the middle and upper stratosphere where the negative (positive) tendency in Eq. (2) is found rather than the positive (negative) tendency found in Eq. (1). It is important to note that the residual term in the TEM equation is shown to be representative of the chemical net production term $\overline{S}$ ($\approx \overline{\chi}_t - \overline{\chi}_{dyn}$). This is an approximation since it also contains ozone transport due to unresolved waves, such as gravity waves (Plumb, 2002). One of the causes for this





discrepancy is that $\overline{S}$ calculations for MERRA-2 rely on the dynamical diagnostic terms in Eq. (2), particularly, the effects of

irreversible eddy mixing transport $e^{z/H}\nabla \cdot M$. The horizontal eddy mixing is predominately influenced by the forcing from the breaking of resolved waves (Plumb, 2002). Furthermore, this discrepancy is very pronounced during winter as shown in Fig. 5f and Fig. 6c. Randel et al. (1994); Minganti et al. (2020) have studied the effect of the SSW event on the N2O TEM budget, which showed more contributions of vertical advection and horizontal eddy mixing to this budget during the SSW event than in the seasonal mean. Thus, we can explain the resulting discrepancy in ozone TEM budget from the highly frequent

occurrence of SSW events in the northern hemisphere, which affects the seasonal cycle in climatology in the polar regions. Hence, the discrepancy in the ozone TEM budget can be accounted for by the more frequent occurrence of midwinter SSW events in the northern hemisphere, leading to the effects on the ozone TEM budget in the polar regions at the seasonal scale. Therefore, determining the ozone transport mechanisms during stratospheric extreme events is one of the keys to improving our understanding of stratospheric processes and ozone variability in stratosphere chemistry-climate models.

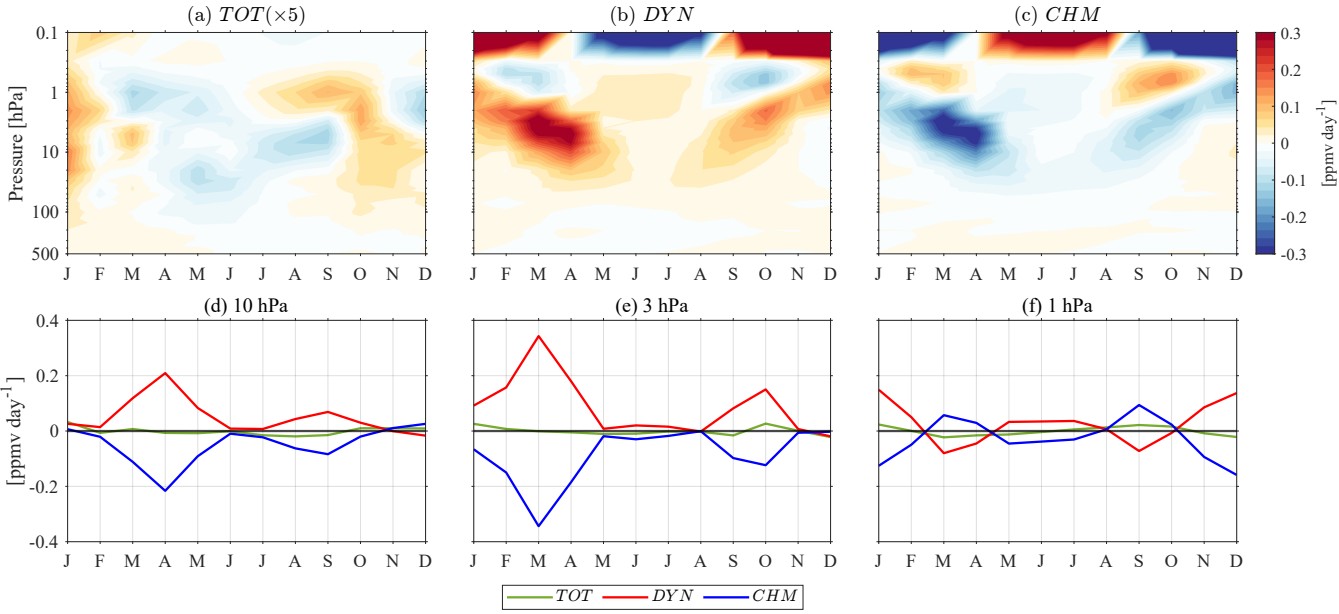

**Figure 6.** Seasonal cycle of (a) total ozone tendency anomaly (TOT), (b) ozone tendency anomaly due to dynamics, and (c) due to parameterized chemistry averaged the polar regions (70 - 90° N) based on Eq. (1). The second row is the comparison of each term at different pressure levels: (d) 10 hPa, (e) 3 hPa, and (f) 1 hPa.

## 5.2 Insights into the Ozone Budget during SSW and FSW events


A climatological comparison of ozone anomalies throughout the life cycle of SSW and FSW events using MERRA-2 and MLS data provides more details about the dynamical and chemical contributions and the temporal evolution of both events. Fig. 7 visualizes the composite vertical structure and evolution of ozone anomalies in MERRA-2 and MLS during SSW and FSW events averaged over the polar regions (70° - 90° N), respectively. The vertically resolved ozone VMR (Fig. 7a, c) shows a




more complicated picture during SSW compared to FSW events (Fig. 7b, d). A weak negative ozone anomaly in the lower and middle stratosphere and a positive ozone anomaly in the upper stratosphere close to the SSW onset are presumably related to the polar stratosphere being dominated by an anomalously strong and cold vortex during this time, leading to reduced transport of ozone rich air masses into the polar regions. Within the first 20 days following the SSW onset, the ozone VMR anomalies rapidly increase by more than 1 ppmv and persist for up to 50 days until late winter in the middle stratosphere (30-10 hPa). The

negative anomalies above 5 hPa exist only shortly at the SSW onset. They are followed by persistent positive anomalies, which tend to reach their maximum value with lags of 20 days at the stratopause and lower mesosphere. During FSW events the ozone anomaly is anomalously negative below the 10 hPa levels before the onset day, exhibiting an about -0.8 ppmv reduced ozone VMR from lags -30 to 0 days compared to the climatology. Above 5 hPa, the negative anomalies persist over the lifetime of FSW and the altitude of the negative anomaly tends to descend with time after the FSW onset. However, the positive ozone

anomalies have a peak in the middle stratosphere at the FSW onset and also persist for about 20 days, propagating downward into the lower stratosphere. The structure of these anomalies differs somewhat from that of SSWs, particularly in the transition from rapidly increasing positive anomalies to descending negative anomalies tendency in the middle-to-upper stratosphere after FSW onset.

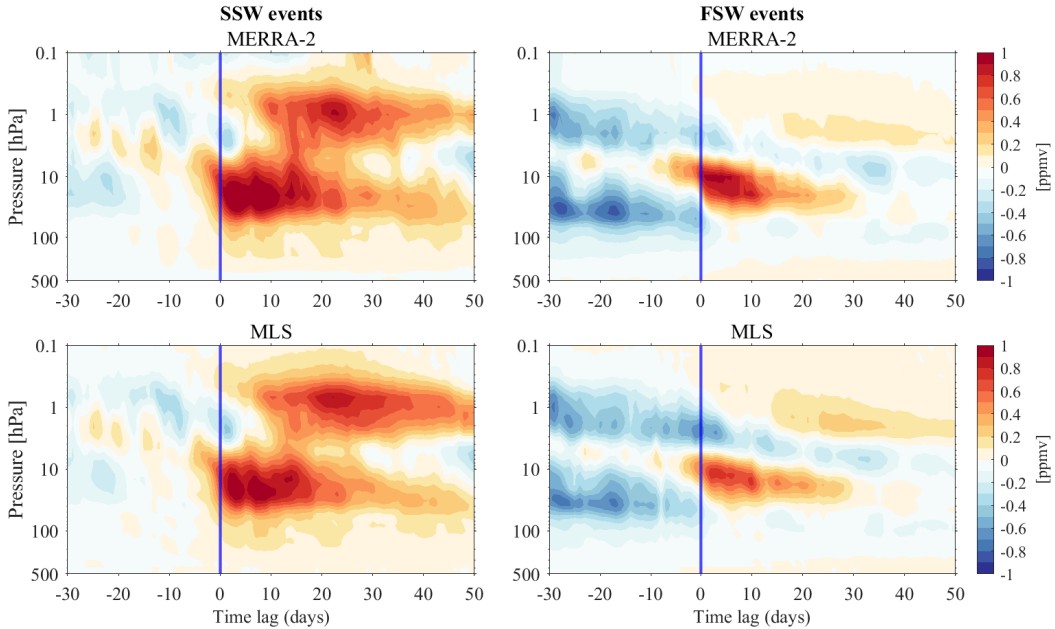

**Figure 7.** Evolution of the ozone anomalies for the composite of SSW and FSW events as a function of time and pressure averaged the polar regions (70° - 90° N) for (a, b) MERRA-2 and (c, d) MLS.

Utilizing the results obtained from vertically integrated ozone tendency and ozone continuity equations, we compare the specific contributions of dynamical and chemical processes to the observed ozone anomaly behavior during SSWs and FSWs.



Fig. 8 and Fig. 9 present the anomalous ozone tendencies averaged between 70° and 90° N during SSW and FSW events, along with their associated dynamical and chemical fields. We have omitted the contributions due to the horizontal advection (not shown) since it is small compared to the other processes. Moreover, we decompose the total ozone tendency into contributions

of dynamical and chemical terms in Eq. (1) to infer the sources of transient changes in the polar stratospheric ozone tendency anomalies.

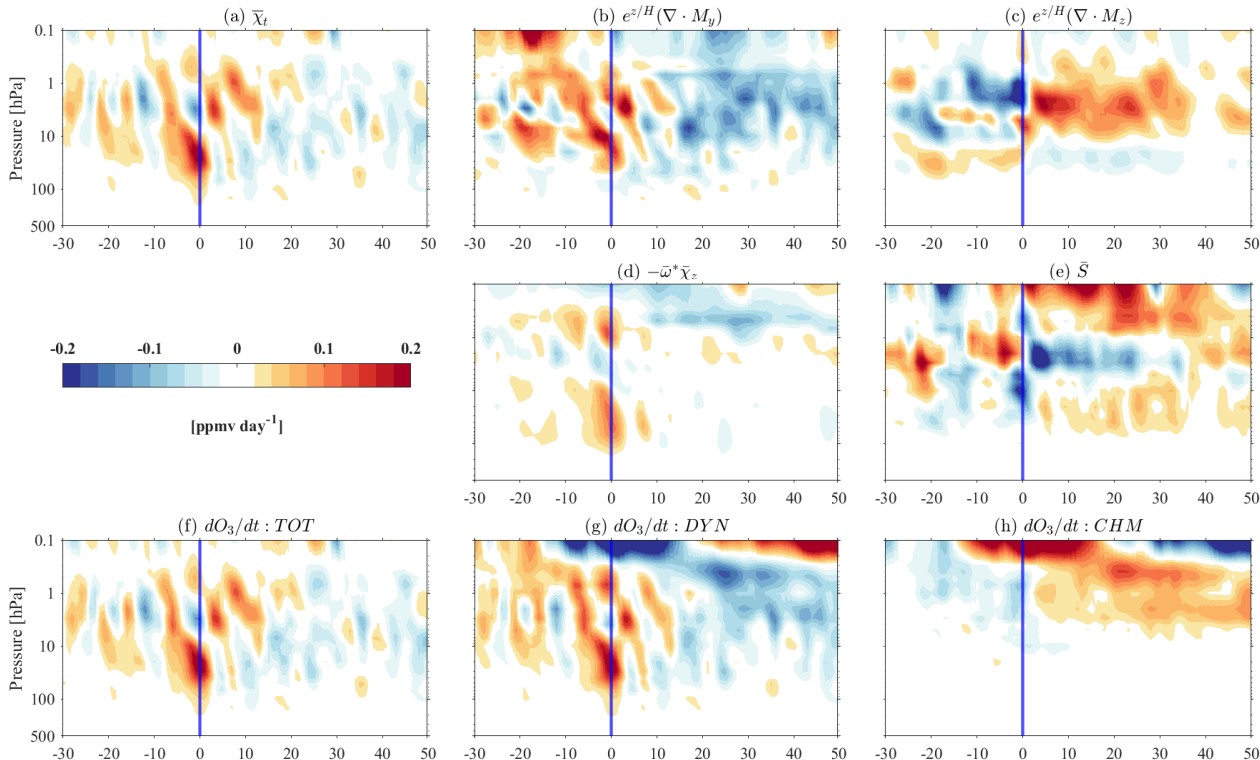

**Figure 8.** Anomalous ozone tendencies for SSW events as a function of time and pressure averaged the polar regions (70° - 90° N) from MERRA-2: calculation of (a) ozone tendency anomaly $\overline{\chi}_t$ both due to the dynamical field $\overline{\chi}_{dyn}$ that is decomposed into (b) horizontal eddy transport effect $e^{z/H}\nabla \cdot M_y$, (c) vertical eddy transport effect $e^{z/H}\nabla \cdot M_z$, (d) vertical advection transport $-\overline{\omega}^*\overline{\chi}_z$, and due to chemical field that is (e) chemical net $\overline{S}$ based on Eq. (2). The third-row shows (f) total ozone tendency anomaly $TOT$, (g) ozone tendency anomaly due to dynamics $DYN$, and (h) due to parameterized chemistry $CHM$ based on Eq. (1).

The results indicate pronounced ozone tendency anomalies $\overline{\chi}_t$ primarily between 100 and 10 hPa, starting with a positive ozone tendency anomaly from lag -8 days to the onset day (Fig. 8a). The evolution of ozone tendency anomaly $\overline{\chi}_t$ is consistent

with $TOT$ anomaly in Fig. 8f. The evolution of $TOT$ anomaly in the lower and middle stratosphere (between 100 and 1 hPa) is mainly dominated by $DYN$ (Fig. 8g). $DYN$ in the lower and middle stratosphere (100 and 10 hPa) is primarily attributed to the ozone transport via vertical advection (Fig. 8d) and eddy transport effects (Fig. 8b, c). Notably, the dominance of the



dynamical term on $\overline{\chi}_t$ or $TOT$ anomaly in the lower stratosphere during the life cycle of the SSW composite is consistent with the transient changes in vertical residual mean transport (Fig. 2). A strong negative $\bar{\omega}^*$ exists in the polar regions corresponding

to the positive $\overline{\chi}_{dyn}$ and $TOT$ (Fig. A1). An intensified residual circulation significantly weakens or breaks up the polar vortex, hence, facilitates poleward ozone transport resulting in an increase of ozone VMR (Schranz et al., 2020; Shi et al., 2023; Bahramvash Shams et al., 2022; Harzer et al., 2023).

In the upper stratosphere, there is a notable negative ozone tendency anomaly at the onset of SSW, primarily driven by horizontal eddy and vertical advection. The vertical eddy effect contributes to building up the negative ozone tendency anomaly,

while the chemistry term $\overline{S}$ tends to compensate for the ozone tendency anomaly from lag -10 days to the SSW onset day. Conversely, as the vertical eddy effect builds up the ozone tendency anomaly, the chemistry term $\overline{S}$ balances/weakens the ozone tendency anomaly after the SSW onset. Positive vertical eddy transport $e^{z/H}\nabla \cdot M_z$ and negative $\overline{S}$ in the upper stratosphere (between 3 and 1 hPa) from SSW onset day 0 to lag 35 days partially counteract the dynamically induced ozone anomalies through $\overline{S}$ (Fig. 8e). Interestingly, Fig. 8h presents two opposite attributions of $CHM$ before and after SSW onset (the pe-

riod is from lag -15 days to 35 days) in the upper stratosphere, respectively, which is almost the opposite tendency compared with $\overline{S}$. The significant discrepancy during SSW events is evident in $\overline{S}$ with negative contributions in the upper stratosphere (Fig. A2), possibly attributed to uncertainties in calculating the eddy transport term, along with uncertainties in the rest of the dynamical terms. A potential source of discrepancy is that Eq. (2) does not account for the effects of numerical diffusion and vertical diffusion due to the gravity wave parameterization, in particular, which are presumably non-negligible in the middle

to the upper stratosphere. As discussed in (Brasseur and Solomon, 2005), the polar middle stratosphere, as the transition layer, is intricate and requires consideration of various conditions and additional constraints. This is because ozone is chemically controlled above this layer, while below it is dynamically controlled.

During FSW, the anomalous ozone tendency $\overline{\chi}_t$ exhibits notable differences compared to SSW events, particularly in the middle stratosphere the FSW ozone tendency is affected by chemistry and dynamical process-induced wave-mean flow interactions.

In Fig. 9, the negative $\overline{\chi}_t$ anomaly lags from 5 to 15 days in the middle stratosphere which is attributed to the photochemical effects counteracted (Fig. 9e, h) partially the positive horizontal eddy transport (Fig. 9b). In the lower stratosphere, the strong anomalous positive tendency $\overline{\chi}_t$ at FSW onset (Fig. 9a) is associated with the dynamical terms which are horizontal eddy and enhanced vertical advection transports (Fig. 9b and Fig. 9d). In the upper stratosphere, there is no obvious ozone tendency anomaly at FSW onset, which can be explained by the negative contributions of vertical eddy transport (Fig. 9c) counteracted

by other terms. The evolution of the $TOT$ and $\overline{S}$ are consistent with $\overline{\chi}_t$ and $CHM$, respectively, in the lower and middle stratosphere (50-3 hPa) during FSW. In addition, the strong $\overline{\chi}_t$ and $CHM$ around the FSW onset emphasize the importance of chemical processes in spring.

There is also a remarkable agreement between $\overline{S}$ and $CHM$ (as well as $\overline{\chi}_{dyn}$ and $DYN$) anomalies in the lower mesosphere during the SSW and FSW events are displayed in Fig. 8d, h. It can be attributed to the temperature-ozone relation that suggests

that in a region dominated by pure oxygen chemistry, a temperature decrease of 10K would produce about a 20% increase in ozone Brasseur and Solomon (2005). Temperature changes will modify all temperature-dependent photochemical rates and hence feedback to the ozone chemistry. As shown in Fig. 1d and Fig. 1h, from lags 10 to 40 days during SSW events in the




lower mesosphere, the temperature negative anomaly is more than 10K from lags 10 to 40 days and the ozone VMR positive anomaly reaches up 0.5 ppmv. Ozone anomaly resulting from the negative dynamical transport and chemical production mani-

fests a few days after SSW onset and lasts for an extended period of 50 days. During FSW events, positive dynamical transport and net chemical loss nearly balance each other at 0.5 hPa (Fig. A2), leading ozone tendency anomalies to fluctuate around zero.

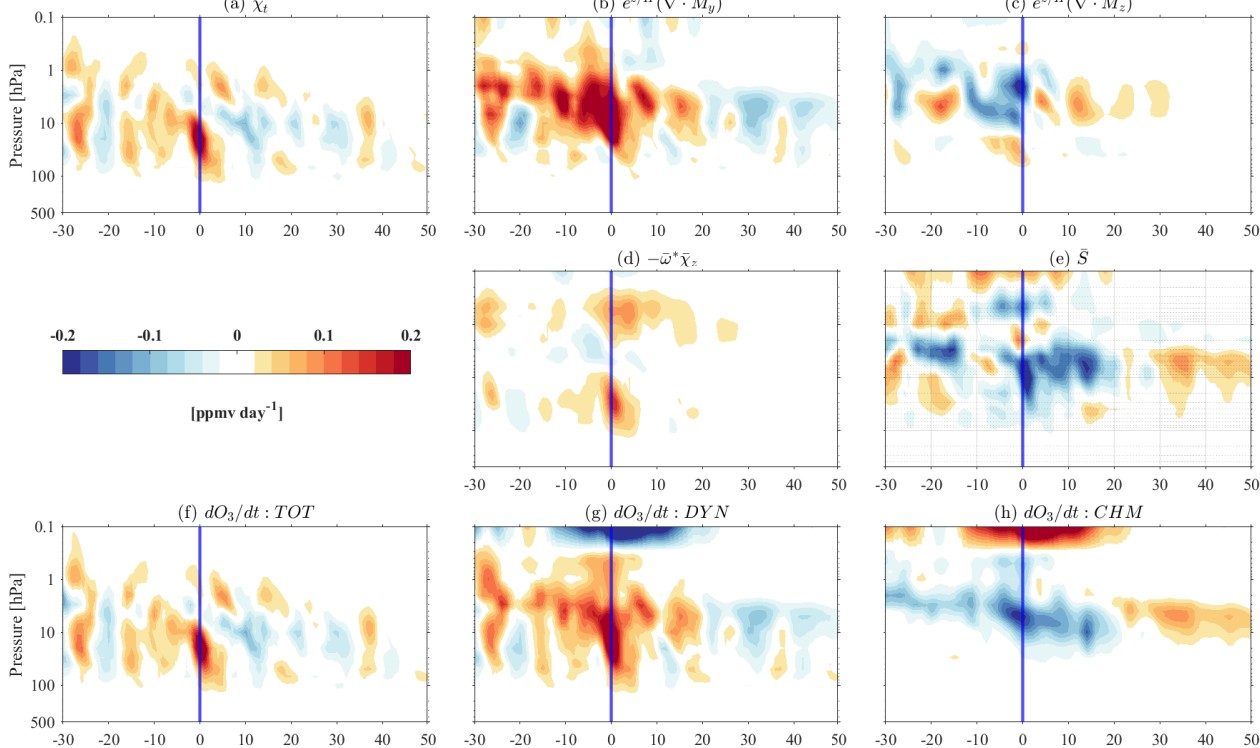

**Figure 9.** As the same in Fig. 8, but for the FSW composite events.

### 5.3 Dynamical Control of Total Column Ozone

Many studies have highlighted the significant impact of enhanced propagation of planetary waves in the lower stratosphere

on the increase of TCO in winter (Matthias et al., 2013; Shaw and Perlwitz, 2014; Lubis et al., 2017; Safieddine et al., 2020; Matthias et al., 2021), subsequently leading to reduced ozone depletion in springtime (Manney et al., 2020; Lawrence et al., 2020; Schranz et al., 2020). The positive TCO anomalies after SSW events spanning a period exceeding 40 days analyzing data from ERA5 and MERRA-2 reanalysis data, MLS, or comprehensive GCMs such as WACCM over the polar regions (de la Cámara et al., 2018; Safieddine et al., 2020; Bahramvash Shams et al., 2022; Hocke et al., 2023). Robust positive TCO

anomalies during early FSW events are influenced by the wave geometry of the FSW (Butler and Domeisen, 2021). Therefore, the dynamical behavior of SSW and FSW events which alter the chemical and dynamical evolution of the polar stratospheric



ozone VMRs affect the distribution of TCO over the northern polar region.

Quantitatively separating the effects of dynamic and chemical processes in TCO is challenging because polar ozone is similarly affected by each process. Therefore, we focus on the variability caused by dynamical processes in the TCO changes based on

the relative contributions of dynamical and chemical processes in section 5.2. We calculate TCO tendency anomalies over polar regions (70 - 90° N) during SSW and FSW events using MERRA-2 in Fig. 10. We find that enhanced polar TCO close to the SSW and FSW onset is mainly induced by anomalous horizontal eddy effect and vertical advection transport in the lower stratosphere (at 30 hPa in Fig. 10). Concerning the 450 K level, it turns out that the involved dynamical processes affect the polar TCO tendency anomalies. Fig. 10 and Fig. A3 indicate the polar TCO anomalies during SSW and FSW events can be

attributed to anomalous dynamical processes.

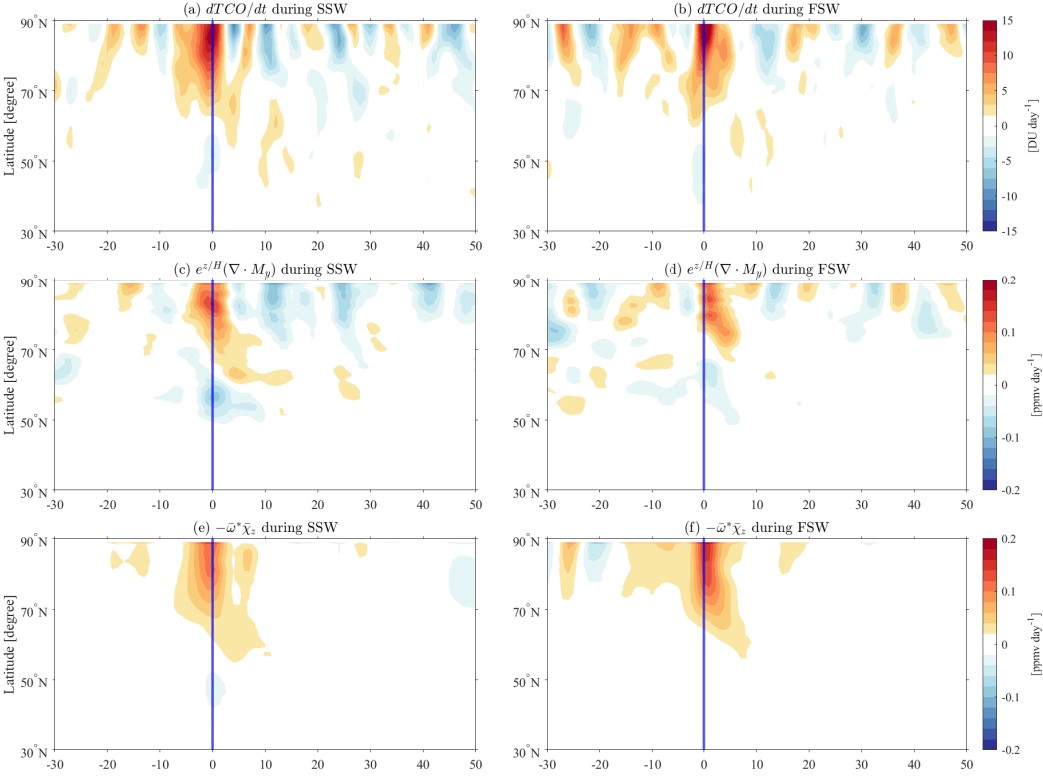

**Figure 10.** Evolution of the TCO tendency anomalies $dO_3/dt$ (DU day$^{-1}$), horizontal eddy effect $e^{z/H}(\nabla \cdot M_y)$, and vertical advection transport $-\bar{\omega}^*\bar{\chi}_z$ anomalies at 30 hPa (ppmv day$^{-1}$) for the composite of SSW (a, c, e) and FSW (b, d, f) events as a function of time and latitude in the northern hemisphere.





## 6 Discussion and Conclusions

In this paper, we use MERRA-2 and MLS data to identify the SSW and FSW events by analyzing zonal wind fields and polar temperatures covering the period from 2004 to 2022. We focused on investigating the vertically resolved polar ozone variations during both SSW and FSW events and quantifying their driving mechanisms. The impact of major SSW and early FSW events

on ozone in the stratosphere and mesosphere was investigated using microwave radiometer measurements GROMOS-C at Ny-Ålesund, Svalbard (79° N, 12° E). Microwave observations of the daily ozone profiles during SSW and FSW events were retrieved in the stratosphere and lower mesosphere at 20-70 km. GROMOS-C captured the high variability of middle stratosphere ozone fluctuations, showing a dramatic increase in ozone VMRs after SSW and FSW onset. For validation purposes, local changes in ozone VMRs from MERRA-2 in the stratosphere and mesosphere displayed common features in GROMOS-C

and MLS under the SSW and FSW conditions. Ozone anomalies are identified throughout the stratosphere and lower mesosphere (from 100 to 0.1 hPa) during SSW and FSW events. Notably, positive ozone VMR anomalies of approximately 1.5 ppmv in the middle stratosphere persisting for 30 days after SSW onset and 20 days after FSW onset have been documented. Based on the TEM budget equation, we rationalize the impact of SSW and FSW events on ozone anomalies by calculating dynamical and chemical terms in Eq. (2) via meteorological variables provided by MERRA-2 reanalysis data:

1. The enhanced transport of ozone into the polar cap on the seasonal scale is attributed to the increased occurrence of SSW events during midwinter in the northern hemisphere. However, more ozone chemical loss in springtime than the climatology of the seasonal mean is attributed to more early FSW events.

2. The impact of SSW and FSW events on total ozone tendency is shown by the altitude tendencies from the lower to middle stratosphere (from middle stratosphere to upper stratosphere and lower mesosphere) that change from positive to
negative (from negative to positive) close to onset.

3. Positive ozone anomalies larger than 1 ppmv close to SSW onset in the lower and middle stratosphere are attributed to the dynamical processes of the horizontal eddy effect and vertical advection transport, while this response pattern for FSW events is associated with the combined effects of dynamical and chemical terms, reflected by the photochemical effect counteracted partially by positive horizontal eddy transport, in particular in the middle stratosphere.

4. Substantial differences in the chemical fields in the upper stratosphere displaying negative $\overline{S}$ and positive $CHM$ after SSW onset within 30 days, are attributed to greater uncertainties in TEM diagnostics, particularly in calculating eddy effects and mean advection transports.

Our results establish a new perspective on the driving mechanisms behind pronounced polar ozone anomalies associated with dynamical and chemical processes in the stratosphere during SSW and FSW events. Although previous studies have shown

composite spatial and temporal ozone response to SSW events in the Arctic (de la Cámara et al., 2018; de la Cámara et al., 2018; Hong and Reichler, 2021; Bahramvash Shams et al., 2022; Harzer et al., 2023), we took a more comprehensive approach and higher altitude up to the lower mesosphere to study the polar ozone anomalies, and considered not only major SSW events



but also early FSW events. The polar ozone response pattern reflects the underlying ozone transport anomalies when viewed over the polar latitude station with a vertically resolved response structure. The ozone response signature during SSW and

FSW events in the stratosphere and lower mesosphere can be explained by consecutive counteracting anomalous tendencies associated with eddy mixing effects and advection transports on daily timescales, as well as chemical production and loss. In particular, these studies showed that weaker midwinter planetary wave forcing in the stratosphere due to weaker upward wave propagation leads to lower spring Arctic temperatures, and thus to more ozone destruction in spring. In particular, our results suggest that anomalous ozone tendency during FSW events in the middle stratosphere can be attributed to the dynamical

field counteracted partially by chemical loss. Furthermore, the type of SSW is characterized by anomalous evolution of ozone tendencies in winter, leading to distinct chemistry patterns and variations in intensity and duration of anomalous transport and mixing properties in the upper stratosphere. In contrast, chemistry contributions during years with FSW events in spring are relatively less pronounced in the upper stratosphere, representing a predominantly smooth transition according to climatology. Finally, referring back to a novel aspect of this study involving the relative contributions of dynamical and chemistry effects

to the anomalous ozone tendency, we found a significant discrepancy in chemical effects between $\overline{S}$ utilizing TEM diagnostic and $CHM$ from chemistry transport models is observed during SSW events, which is not replicated in FSW events, as shown in Fig. 9e, h. This finding contributes to a growing body of evidence suggesting that the difference is associated with the substantial uncertainties in the calculated dynamical terms derived from the MERRA-2 reanalysis for SSW events. However, it is unclear whether the remaining differences only result from the quality of the reanalysis data, and substantial anthropogenic

ozone-depleting substances in recent decades, indicating that ozone chemistry has become increasingly important in governing climate variability (Sun et al., 2014; Banerjee et al., 2020; Schranz et al., 2020; Shi et al., 2023). In addition, we found that the ozone tendency in the lower stratosphere is primarily attributed to the horizontal eddy effect and vertical advection transport. Thus, we consider the observed variability in zonal-averaged TCO in the polar regions for SSW and FSW events from MERRA-2. Dynamical processes in the lower stratosphere dominate TCO variability.

In general, the findings of this study contribute to a more comprehensive interpretation of the observed ozone variability at polar stations, with particular emphasis on the ozone anomaly situation. While existing research has predominantly concentrated on dynamic effects on Arctic ozone (de la Cámara et al., 2018; Bahramvash Shams et al., 2022; Harzer et al., 2023), our study emphasizes the combined contribution of dynamical and chemical effects in polar ozone anomalies. It is especially evident that the anomalies of polar TCO during SSW and FSW events can be attributed to wave-driven anomalous dynamics. Therefore,

understanding the interplay between dynamical and chemical processes during stratospheric extreme events will enhance our comprehension of the connections between middle and upper stratospheric dynamics and ozone chemistry. This knowledge is crucial for interpreting the observed vertically resolved pattern of daily variability and better quantifying polar ozone evolution.

*Data availability.* The GROMOS-C and MIAWARA-C level 2 data are provided by the Network for the Detection of Atmospheric Composition Change and are available at http://www.ndacc.org (NDACC, 2022). MLS v5 data are available from the NASA Goddard Space

Flight Center Earth Sciences Data and Information Services Center (GES DISC): https://doi.org/10.5067/Aura/MLS/DATA2516. MERRA-



2 data are provided by NASA at the Modeling and Assimilation Data and Information Services Center (MDISC) and are available in the model level (GMAO, 2015a) at https://doi.org/10.5067/WWQSXQ8IVFW8 and in pressure level (GMAO, 2015b) at https://doi.org/10.5067/QBZ6MG944HW0 and ozone tendency at https://doi.org/10.5067/S0LYTK57786Z.

## Appendix A



**Figure A1.** Evolution of the $\bar{\omega}^*$ anomalies $dO_3/dt$ (m s$^{-1}$), $\overline{\chi}_{dyn}$, and $DYN$ anomalies at 30 hPa (ppmv day$^{-1}$) for the composite of SSW (a, c, e) and FSW (b, d, f) events as a function of time and latitude in the northern hemisphere.



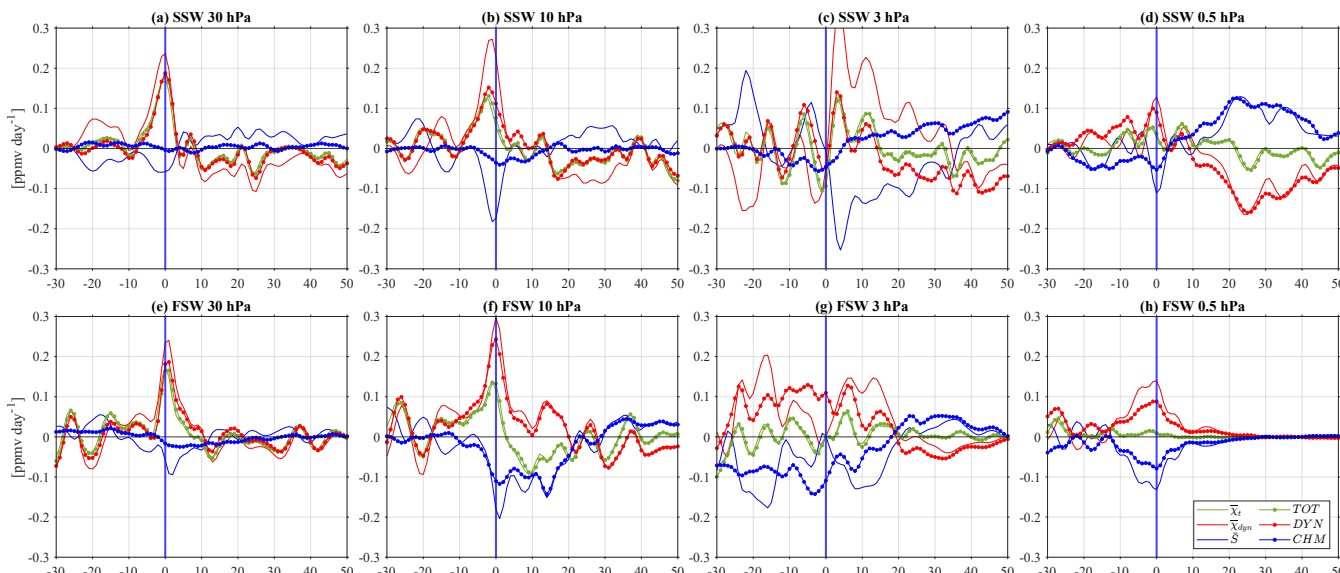

**Figure A2.** Comparision the composite evolution of the anomalies of ozone tendency, dynamical term and chemical term between Eq. (1) and Eq. (2) at 30 hPa, 10 hPa, 3 hPa, and 0.5 hPa, averaged over $70°$ - $90°$ N from MERRA-2, (a, b, c, d) for SSW events and (e, f, g, h) for FSW events.



**Figure A3.** Same as Fig. 10, but for pressure level at 20 hPa.

*Author contributions.* GShi was responsible for the ground-based ozone measurements with GROMOS-C, performed the data analysis, and prepared the manuscript. ES provided the Aura-MLS data. WK helped with data analysis. GStober designed the concept of the study and contributed to the interpretation of the results. All of the authors discussed the scientific findings and provided valuable feedback for manuscript editing.

*Competing interests.* The contact author has declared that none of the authors has any competing interests.



*Acknowledgements.* Guochun Shi, Gunter Stober, and Witali Krochin are members of the Oeschger Center for Climate Change Research. The authors acknowledge NASA Global Modeling and Assimilation Office (GMAO) for providing the MERRA-2 reanalysis data and the Aura/MLS team for providing the satellite data.

*Financial support.* This research has been supported by the Swiss National Science Foundation (grant no. 200021-200517/1).



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
