# Peer review of "Ozone anomalies over polar regions during the stratospheric warming events"

_EGUsphere, 2024_

## Author Comment (AC1)

**Response to referee#1's comments for manuscript**

The authors would like to thank the substantial comments and suggestions from the referees, which significantly helped improve the quality of this manuscript. We have revised the manuscript carefully based on the comments and suggestions of the reviewer. More details of the revision can be found in the revised manuscript as well as the point-to-point response as follows (all authors' responses here are in blue).

In this paper, ozone changes in the middle atmosphere around sudden stratospheric warming (SSW) and early final stratospheric warming (FSW) events are analyzed. MERRA-2 data are used to disentangle the different dynamical and chemical processes affecting ozone in the stratosphere and mesosphere. MERRA-2 data are evaluated against observations of GROMOS-C and MLS, and the good qualitative agreement in the ozone anomalies between GROMOS-C, MLS, and MERRA-2 below 0.1 hPa justify the use of MERRA-2 for the analysis of dynamical versus chemical processes. This is an interesting analysis, and the approach used to disentangle advection, turbulence, and chemistry is to my knowledge novel. In this sense, the paper brings new aspects to our understanding of these large disturbances of middle atmosphere dynamics; as SSWs are know to affect the whole atmosphere from the troposphere to the ionosphere, understanding its drivers and impacts is of course very important. However, I found the paper in parts difficult to follow. In particular how ozone is implemented in MERRA-2, and how it depends on the different chemical and dynamical terms is unclear (Section 2.3). As this is a prerequisite to understand the analysis and interpret the results, this should be clarified, and I have listed more specific comments to this issue as Major points below. I also have a rather long list of minor comments mostly regarding unclear wording listed below.

**Major comments:**

1. Lines 111-112: in the stratosphere, odd oxygen is dominated by ozone, but in the upper mesosphere, it is dominated by atomic oxygen. This has to be taken into account when comparing MERRA-2 ozone to GROMOS-C and MLS, and presumably can explain the much higher values of MERRA-2 in the upper mesosphere.

   We completely agree that this distinction in atmospheric composition between the stratosphere and upper mesosphere must be carefully considered when interpreting and comparing ozone data from different sources. The revised manuscript is going to add this information about MERRA2 ozone in the model section.

2. Lines 111-114: I don't understand the relationship between the different models here. As I understand your description, MERRA-2 uses tendencies of Ox (not ozone) from the GEOS CTM as ozone tendencies. This would explain the very high ozone values of ozone in the upper mesosphere, as Ox (and Ox tendencies) is significantly higher there, than ozone. But why are the assimilated meteorological data from GMAO used here, not from MERRA-2? I presume that you mean that these are used within the GEOS-CTM, not within MERRA-2, but this is not clear. Also, if the GMAO meteorological data are used to derive the ozone tendencies than used in MERRA-2, that would imply that the resulting ozone fields in MERRA-2 are inconsistent with dynamics of MERRA-2. I don't think this is the case, but can you please clarify?

   Ozone tendency: these datasets are assimilated with the Goddard Earth Observing System Model, version 5 (GEOS-5) by using odd-oxygen mixing ratio as its prognostic variable. This includes an odd-oxygen family transport model that provides the ozone concentration necessary for solar absorption. The vertically integrated ozone tendency is given as following equation. In addition, the revised manuscript cites two publications that use the ozone tendency from MERRA-2 to investigate the polar stratospheric ozone and provide a description of the datasets.

   Lubis, S. W., Silverman, V., Matthes, K., Harnik, N., Omrani, N.-E., and Wahl, S.: How does downward planetary wave coupling affect polar stratospheric ozone in the Arctic winter stratosphere?, Atmospheric Chemistry and Physics, 17, 2437–2458, 2017.

   Bosilovich, M. G.: MERRA-2: Initial evaluation of the climate, National Aeronautics and Space Administration, Goddard Space Flight Center, 2015. https://gmao.gsfc.nasa.gov/pubs/docs/Bosilovich785.pdf

3. Line 115: what is vertically integrated here – ozone, or the ozone tendency? Vertically integrated ozone would be total ozone; as you are discussing ozone profiles here, is it possible that you mean "vertically resolved", not

"vertically integrated"?

The model uses an odd oxygen mixing ratio, as its prognostic variable. It is the vertically integrated ozone tendency.

4. Line 117: shouldn't the left-hand side be the total derivative, not a partial derivative?

The left-hand side is a partial derivative. https://gmao.gsfc.nasa.gov/pubs/docs/Bosilovich785.pdf

5. Line 121: can you explain in a bit more detail how ANA refers to assimilation of ozone? Also, is this done within MERRA-2, or within the GEOS CTM?

Ozone is analyzed and so can change due to ozone increments or due to increments in atmospheric dry mass. We report only the total analysis contribution. To compare the tendencies with the total tendency from the states, a conversion factor of 1.65 mol/mol must be applied to the tendencies. https://gmao.gsfc.nasa.gov/pubs/docs/Bosilovich785.pdf

6. Line 189 and Section 4: in my view, the value of the GROMOS-C (and MLS) data is that the evaluation of the performance of MERRA-2 ozone in relation to the FSW/SSW events; the good qualitative agreement is the justification to use MERRA-2 to analyze the dynamics versus chemistry in a later step. This should be made very clear here.

We will add this justification as suggested by the reviewer: In this section, our analysis reveals a qualitative agreement between MERRA-2 ozone data and observations from GROMOS-C and MLS instruments during FSW/SSW events. This agreement serves as a robust justification for employing MERRA-2 data to explore the dynamics versus chemistry relationship in subsequent steps of our research, providing confidence in the reliability of MERRA-2 ozone data for our analytical purposes.

7. Line 198: If I understood correctly how ozone is implemented in MERRA-2 (description in Sec. 2.3), this is Ox, not O3; in the stratosphere and lowermost mesosphere, the difference is negligible, but in the upper mesosphere, there is significantly more Ox than O3 – this presumably explains the very high values of MERRA-2 "O3" shown in Figure 3 c) and f). If this is correct, MERRA-2 ozone can not be used above 0.1 hPa. Please clarify.

Changed: The results indicate a good agreement between MERRA-2 (below 0.1 hPa) and MLS with GROMOS-C observations. However, due to the complexity of altered dynamics in the winter polar regions introducing extra uncertainties into numerical models and data assimilation systems (Wargan et al., 2017), ozone VMRs exhibit dramatic variability (in Fig.3b, e) in the mesosphere from MERRA-2.

8. Lines 374 and following, Discussion and conclusions: Please be more precise which data you used, and what for. You used MERRA-2 temperatures and wind fields to identify SSWs. You probably did not use MLS wind fields for this as stated here, as MLS does not observe winds(?). You used MLS and GROMOS-C ozone to evaluate MERRA-2 ozone fields, and ozone anomalies, and justify its use for analyzing the dynamical and chemical drivers of the ozone changes. Please clarify this here.

Changed: In this paper, we use MERRA-2 reanalysis data to identify the SSW and FSW events by analyzing zonal wind fields and polar temperatures covering the period from 2004 to 2022. A qualitative agreement in ozone between MERRA-2 and observations from GROMOS-C and MLS instruments during FSW/SSW events provides confidence in the reliability of MERRA-2 data to investigate the driving mechanisms of polar ozone dynamics and chemistry.

**Minor comments:**

1. Lines 13-15: it is not clear what "this response pattern" refers to here. Maybe better "FSW events are associated with …"

Changed: The pattern of ozone anomalies for FSW events is associated with the combined effects of dynamical and chemical terms, which reflect the photochemical processes counteracted partially by positive horizontal eddy transport, in particular in the middle stratosphere.

2. Line 16: which chemistry-climate model? In the paper, model results are shown from MERRA-2 whose ozone product is based on a chemistry-transport model. No chemistry-climate model results are shown.

   Changed: Here, we contrast results from the ozone continuity equation using MERRA-2 reanalysis data and direct ozone tendency based on the odd-oxygen family transport model to quantify the impact of dynamical and chemical processes on ozone anomalies during SSW and FSW events.

3. Line 15-17: this sentence should come after the sentence ending in line 10, to clarify where the results discussed in lines 7 and following come from. Else it is not clear where "the underlying dynamical and chemical mechanism" discussed in the sentence starting in line 10 comes from.

   Changed

4. Line 26: SSW events (plural)

   Changed

5. Line 29: Observed FSW events . . . . depend (plural)

   Changed

6. Line 31: atmospheric species, not atmosphere species

   Changed

7. Lines 32-33: ". . . . ozone plays the most important role in the coupling between chemistry, radiation, and dynamical processes in the stratosphere and lower mesosphere. " Ozone radiative heating and cooling peaks at the stratopause; in the upper mesosphere, heating by O2 becomes important as well.

   Changed: ozone plays the most important role in the coupling between chemistry, radiation, and dynamical processes in the stratosphere and lower mesosphere. Ozone radiative heating and cooling peaks at the stratopause, in the upper mesosphere, heating by oxygen becomes important as well.

8. Line 23-45: There are less studies of the impact of SSWs on mesospheric ozone, but there is some literature about this as well, e.g., Tweedy et al., JGR, 2023; Smith-Johnsen et al., JASTP, 2018. These should be discussed here as well.

   Added: Tweedy et al. (2013) use output from SD-WACCM to explore the evolution of secondary ozone in the mesosphere during SSWs associated with anomalous vertical residual motion and consistent with photochemical equilibrium governing the mesosphere-lower thermosphere (MLT) nighttime ozone. Smith-Johnsen et al. (2018) investigates the cause of the mesospheric nighttime ozone increase during the 2002 Southern Hemisphere winter which is attributed largely to enhanced upwelling and the associated cooling of the altitude region in conjunction with the wind reversal.

9. Line 61: In addition, we show . . .

   Changed

10. Line 61-63: this is stating the obvious – as total ozone is dominated by the amount of ozone in the lower stratosphere, anything affecting lower stratosphere ozone will have a correlating response in total ozone.

    In addition, we show that polar ozone anomalies in the lower stratosphere mainly predominantly governed by the horizontal eddy effect and vertical advection transport processes exhibit a strong correlation with polar total column ozone corresponding to both types of events.

11. Line 75: using instead of leveraging

   Changed

12. Line 80: and instead of which

   Changed

13. Line 80-82: are you using the same retrieval and calibration version as in Fernandez et al (2015)?

   Yes, we use the same retrieval and calibration version for the GROMOS-C.

14. Line 84: instrument, not instruments (?)

   Changed: instrument

15. Line 88-89: please clarify what depends on the pressure here – the ozone profile, not the 240 GHz microwave band

   Changed: The ozone profile is retrieved using the 240 GHz microwave band, which extends from 261 hPa to 0.0215 hPa for recommended scientific applications.

16. Line 93: available instead of applicable

   Changed

17. Line 135: X(dyn) consists of three terms, not four

   X(dyn) consists of four terms: the horizontal and vertical advection, horizontal and vertical eddy transport effects

18. Line 139: in this equation, one bracket is missing, probably at the end

   Changed

19. Lines 164-165: I would say the westerly starts to weaken in 10-.1 hPa already a few days before the warming. During the warming, the wind reverses quickly to easterly, and stays like this for at least 30 days below 1 hPa, but reverses back above that.

   Below approximately 0.1 hPa, the westerly wind rapidly weakens lags - 10 days and then switches to an easterly wind after the SSW onset (lags 0 days) at 10 hPa in Fig.1a. The easterly wind stays like this for at least 30 days below 1 hPa but reverses back above that.

20. Line 165-166: I think you mean the westerly winds return after approximately 15 days? However, only above 1 hPa

   Changed: The easterly wind returns after approximately 15 days at 10 hPa. After around 20 days of SSW onset, wind at 0.1 hPa reverses to westerly with a maximum speed of 80 m/s and stays like this for at least 20 days.

21. Line 168: .... and cooling in the mesosphere above 0.1 hPa, which seems to onset a few days before the warming?

   Changed: The temperature fields undergo alterations in conjunction with the wind field weakening.

22. Line 168: In Fig. 1b) ... please state here that now you are discussing FSWs, not SSWs. Note that the westerlies begin to weaken at lag -10 as well (similar to SSWs), and even reverse above 0.1 hPa before the event.

   In Fig.1b, the zonal-mean zonal wind at 60°N and 10 hPa during the FSW event is easterly with lags 50 days until the early summer and does not reverse to westerly. We emphasize the reversal of wind at 10 hPa.

23. Line 170: at the stratopause. Temperatures in the lower stratosphere also increase strongly, but there is cooling in the upper mesosphere.

    Added: Temperatures in the lower stratosphere also increase strongly, but there is cooling in the upper mesosphere.

24. Line 173: the mean climatology of all years, or of all years without SSWs/FSWs? If SSW years are included in the climatology, that will diminish the anomalies somewhat.

    The mean climatology is all years with SSW/FSW events. Although SSW years are included in the climatology, still appear the anomalies somewhat due to the high occurrence of SSW in the northern hemisphere.

25. Line 174: as shown in Fig. 2.

    Changed

26. Line 181: in Fig. 2a?

    Changed

27. Line 183: what does it mean that you have significant anomalies in w* extending to lag -30 before FSWs – those winters are significantly different to other winters much earlier?

    FSWs evolve relatively slowly and result from the sustained lack of stratospheric wave driving, leading to the gradual strengthening and cooling of the vortex. However, the small wave driving sometimes exists starting several weeks before FSW onset such as the minor warming in the winters. This is different from SSWs, as the wave driving during SSW changes much more abruptly during onset. Yes, it means that those winters are different to other winters much earlier.

28. Line 187-188: the lasting w* anomalies after the FSWs at and below 1 hPa are very small though

    Changed

29. Line 196-197: in which altitude range?

    Changed: The results indicate a good agreement between MERRA-2 (below 0.1 hPa) and MLS with GROMOS-C observations.

30. Line 204: erase "with" before descending downward

    Changed

31. Lines 205-206: ... before the FSW onset, which is stronger than before the onset of the SSW events.

    Changed

32. Line 211: Climatological for all years, or for only those without SSW/FSW events? Please clarify.

    Changed: Seasonal changes in ozone tendencies from the eddy effect, advection transport, and chemical loss and production processes based on MERRA-2 reanalysis data for the period 2004-2021 are shown in Fig.5.

33. Line 251-252: how does that explain the difference between 5 and 6?

    We mainly compare the difference between $\overline{\chi}_t$ and $TOT$, $\overline{S}$ and $CHM$.

34. Line 252-254: I agree that it is important to understand the interplay between dynamics and chemistry, which is particularly difficult during strong disturbances of the atmospheric dynamics like SSWs and FSWs. Still, I think the wording here "one of the keys to improving our understanding" is too strong. I would argue instead that SSWs are periods of known stratosphere-troposphere coupling, and that a better understanding of SSWs, and better representation in chemistry-climate models, therefore has the potential to improve medium-range weather forecasts during high-latitude winter.

    Changed: Determining the ozone transport mechanisms during stratospheric extreme events is a better understanding of stratospheric processes and ozone variability in stratosphere chemistry-climate models, and better representation in chemistry-climate models, therefore has the potential to improve medium-range weather forecasts during high-latitude winter.

35. Figure 7: I would say this figure shows essentially the same behavior as 3, though with less noise due to the better sampling; the figure and the discussion of it, are not really necessary, as they repeat things already discussed. In my view, the main use of Figures 3 and 4 is to justify the use of MERRA-2 data for the analysis of dynamics versus chemistry; it is not necessary to do this again.

    Thank you for your perspective on the similarity in behavior between Figure 7 and Figure 4. Figures 3 and 4 primarily illustrate ozone variability at a single polar station and its comparison with GROMOS-C and MLS datasets. Figure 7 serves a distinct purpose. Figure 7 presents ozone anomalies in the polar regions (70-90N) to correspond with the dynamic and chemical processes discussed in the subsequent section. It aims to provide insight into anomalous ozone tendencies during both types of events, providing a broader perspective on ozone behavior across the polar regions during these events. In summary, Figure 7 complements Figures 3 and 4 by focusing on ozone anomalies in the broader polar regions, facilitating a deeper analysis of dynamic and chemical processes influencing ozone variability during specific events.

36. Figure 8: here you use the absolute derivative for TOT, DYN and CHM. This is not consistent with the notation in equation 1, where all are given as partial derivatives. From the setup of equation 1, I think the correct use would be to denote TOT with total derivatives, DYN and CHM with partial derivatives; anyway this should be done in a consistent way throughout the manuscript.

    Thanks for your valuable comment. We have fixed the denotation in Figure 8.

37. Lines 332-334: the sentence is missing a verb.

    Changed: The positive TCO anomalies after SSW events span a period exceeding 40 days analyzing data from ERA5 and MERRA-2 reanalysis data, MLS, or comprehensive GCMs such as WACCM over the polar regions

38. Lines 336-337: as TCO is dominated by the lower stratosphere, changes in lower stratosphere ozone will map directly into TCO.

    Added: As TCO is dominated by the lower stratosphere, changes in lower stratosphere ozone will map directly into TCO.

39. Line 341: 30-90°N, not 60-90°N.

    Changed: We calculate TCO tendency anomalies in the northern hemisphere (30 - 90∘ N) during SSW and FSW events using MERRA-2 in Fig. 10.

---

## Author Comment (AC2)

**Response to referee#3's comments for manuscript**

The authors would like to thank the substantial comments and suggestions from the referees, which significantly helped improve the quality of this manuscript. We have revised the manuscript carefully based on the comments and suggestions of the reviewer. More details of the revision can be found in the revised manuscript as well as the point-to-point response as follows (all authors' responses here are in blue).

**Major comments:**

1. The latitude ranges used are confusing. According to the text, the zonal wind in Figure 1 is at 60N but the omega* anomalies and temperatures are 70-90N. Wouldn't it therefore be more sensible to show the zonal wind at 70N. Can the authors can provide a reason why they did not do this?

   We use the zonal wind at 60°N based on the definition of the major SSW and early FSW: zonal-mean zonal wind at 10 hPa and 60°N (Christiansen 2001; Butler 2015; Baldwin 2021). By displaying the zonal wind data at 60°N, the aim is to capture the broader-scale circulation features, including the subtropical and mid-latitude regions, which can significantly influence the dynamics of the polar vortex and planetary wave propagation. In our study, we focused on the polar regions, therefore, we show the temperature and omega* anomalies inside the polar cap between 70-90°N.

   (a) Christiansen, B., 2001: Downward propagation of zonal mean zonal wind anomalies from the stratosphere to the troposphere: Model and reanalysis. J. Geophys. Res., 106, 27307–27322, doi:10.1029/2000JD000214.

   (b) Butler, A. H., Seidel, D. J., Hardiman, S. C., Butchart, N., Birner, T., and Match, A.: Defining Sudden Stratospheric Warmings, Bulletin of the American Meteorological Society, 96, 1913 – 1928, https://doi.org/https://doi.org/10.1175/BAMS-D-13-00173.1, 2015

   (c) Baldwin, M. P., Ayarzagüena, B., Birner, T., Butchart, N., Butler, A. H., Charlton-Perez, A. J., Domeisen, D. I., Garfinkel, C. I., Garny, H., Gerber, E. P., et al.: Sudden stratospheric warmings, Reviews of Geophysics, 59, e2020RG000 708, 2021.

   (d) Butler, A. H. and Domeisen, D. I.: The wave geometry of final stratospheric warming events, Weather and Climate Dynamics, 2, 453–474, 2021.

2. Ideally I would have thought that his study would be done using some type of coordinate relative to the vortex edge, but perhaps this is difficult in the mesosphere. However, given that the authors have chosen 70N, it would be useful to have some indication of what fraction of the 70-90N area is inside-the-vortex (at least at levels where the vortex can be defined) at the time of the FSW and SSW events. Perhaps the answer is "almost all of it". If this is the case please state this.

   In our study, we mainly focused on the ozone anomalies over the polar regions 70-90 °N after SSW and FSW events. Especially, for SSW, the polar vortex is split or displaced by the planetary waves as shown in Fig. 1 for 2018, 2019, and 2021 SSW events. As suggested by the reviewer most of the volume is inside the polar vortex until the SSW or the FSW events. During these events, it is no longer feasible to define a polar vortex until it recovers.

[Figure]

FIG. 1. Ozone volume mixing ratio variability at 10 hPa from ERA5. The edge of the polar vortex and the location of the Ny-Alesund, Svalbard.

**Minor comments:**

1. Line 194 – "The main benefit of the ground-based observations is the much higher temporal resolution of two hours, which permits to estimate of the sampling bias from the satellite MLS taking data only at two local times." There is no discussion anywhere else in the study suggesting that MLS sampling bias is a problem for this study, so please either delete this sentence or explain why it is relevant.

   Ozone shows a distinct diurnal cycle with up to 60% change depending on local time at the middle/upper stratosphere and lower mesosphere (Schranz et al., 2018), which varies with season. MLS samples at fixed local time and, thus, will always measure ozone at a certain time within this diurnal cycle. That's why we compare GROMOS-C and zonal mean MLS ozone observations.

   Schranz, F., Fernandez, S., Kämpfer, N., and Palm, M.: Diurnal variation in middle-atmospheric ozone observed by ground-based microwave radiometry at Ny-Ålesund over 1 year, Atmos. Chem. Phys., 18, 4113–4130, https://doi.org/10.5194/acp-18-4113-2018, 2018.

2. Line 198 says "The results indicate a good agreement between MERRA-2 and MLS with GROMOS-C observations.", yet in figure 3 – MERRA-2 ozone at altitudes above 0.1 hPa is clearly not in agreement with MLS and GROMOS data.

   Changed: The results indicate a good agreement between MERRA-2 (below 0.1 hPa) and MLS with GROMOS-C observations. However, due to the complexity of altered dynamics in the winter polar regions introducing additional uncertainties into numerical models and data assimilation systems (Wargan et al., 2017), ozone VMRs exhibit dramatic variability (in Fig. 3b, e) in the mesosphere from MERRA-2.

   Wargan, K., Labow, G., Frith, S., Pawson, S., Livesey, N., and Partyka, G.: Evaluation of the ozone fields in NASA's MERRA-2 reanalysis, Journal of Climate, 30, 2961–2988, https://doi.org/10.1175/JCLI-D-16-0699.1, 2017.

3. Figure 4 – Given the MERRA-2 ozone values shown in Figure 3, it does not seem sensible to show these ozone anomalies from 0.1 to 0.01 hPa in Figure 4.

MERRA2 ozone values are provided up to the altitude level of 0.01 hPa. It is true that the ozone volume mixing ratio in MERRA2 seems unrealistic. We explicitly point that in the revision of the MERRA2 plots and remove these altitudes between 0.1 and 0.01 hPa in the anomaly Figures. Due to the complexity of altered dynamics in the winter polar regions introducing extra uncertainties into numerical models and data assimilation systems (Wargan et al., 2017), ozone VMRs exhibit dramatic variability (in Fig. 3b, e) in the mesosphere from MERRA-2.

4. Line 360 - The authors claim an increased occurrence of SSW events during midwinter in the NH. This is not shown or referenced anywhere else in the paper. The statement regarding early FSW events is similarly problematic.

   Between 2003 and 2022 about 10 major SSW events occurred in the northern hemisphere, whereas only 1 event was reported in September 2019 Antarctic SSW in the southern hemisphere within the same period. The total number of events also depends on the methodology to classify SSW events (see reply above). The FSW events were compared to the classification presented by Matthias et al., 2021.

   Matthias, V., Stober, G., Kozlovsky, A., Lester, M., Belova, E., Kero, J. (2021). Vertical structure of the Arctic spring transition in the middle atmosphere. Journal of Geophysical Research: Atmospheres, 126, e2020JD034353. https://doi.org/10.1029/2020JD034353

5. Line 395 – It is not clear what point this sentence is trying to make. The claim that "ozone chemistry has become increasingly important in governing climate variability" certainly needs some justification that is not to be found here.

   There have been several studies showing that the polar vortex dynamics are key to understanding polar ozone VMR (Sun et al., 2014; Banerjee et al., 2020; Schranz et al., 2020, Shi et al., 2023). Due to the ban of chlorofluorocarbons (CFCs) in the Montreal protocol ozone depletion was supposed to stop, and a trend reversal in the circulation is expected. Recent studies show such a trend reversal; however, it is not yet confirmed whether the ozone recovery or the increased carbon dioxide is causal for the changes in dynamics. Monitoring ozone in the stratosphere and lower mesosphere remains therefore a high priority and is supported by the Global Atmospheric Watch Programm (GAW). We will add the references and some explanations.

---

## Author Response (AR2)

We thank the reviewer for the comments and suggestions and revised the manuscript accordingly. All changes are indicated with Latexdiff. Below, we added a point-by-point response to the major concerns. Some of the major concerns are related to the publications of Lubis et al., 2017 and Bosilovich et al., 2015. The later includes a more generic form of equation 1. Our nomenclature and also some statements that are of concern were introduced in Lubis et al. 2017. We rephrased the paragraphs and added additional explanation.

**Major**                                                  **issues:**

**Comment:**
Lines 109-112: the upper limit of the vertical ranges of the two ozone data-sets assimilated into MERRA-2 is of at least as much interest here as the lower limit, as this should limit the usability of the MERRA-2 ozone to higher altitudes. So please state these upper limits as well. To the best of my knowledge, MLS ozone is assimilated up to 0.02 hPa, so the rather large disagreement between MLS and MERRA-2 ozone above 0.1 hPa is a bit strange. You can't resolve this issue in this paper, but you should discuss it in some way.

**Reply:**

We added the upper assimilation limit in the bracket. The differences above 0.1 hPa are explicitly mentioned in the discussion section.

**Comment:**
Line 124: As I already pointed out in my previous review, in the upper mesosphere, odd oxygen is dominated by atomic oxygen, not by ozone; this means that in the upper mesosphere, MERRA-2 "ozone" is actually atomic oxygen, not ozone. As there is much more atomic oxygen in the upper mesosphere than ozone, this could explain the large discrepancy above 0.1 hPa observed compared to MLS and GROMOS ozone. Again, this is not an issue that you can resolve, but you should discuss it.

**Reply:**

We explicitly added the suggested discussion to the MERRA2 section and expanded the description.

**Comment:**
Line 124: I am not quite sure the term "prognostic variable" is used correctly here. As I understood equation (1) in this context, the production and loss terms of the assimilation model are derived for odd oxygen, not for ozone. As particularly the chemical production and loss terms are much simpler for odd oxygen than for ozone, this approach makes sense, particularly in the stratosphere where odd oxygen is dominated by ozone.

**Reply:**
The term prognostic was introduced in Ludis et al 2017 (page 2439, first paragraph left column (top)). However, we understand the concern of the reviewer and have changed the name to 'diagnostic'. The odd-oxygen model is used to estimate the state of the ozone rather its temporal evolution or a future prediction in the context of the data assimilation.

**Comment:**
Lines 127-129: again, I am not clear whether "vertically integrated" is the correct term here, either for ozone itself, or the ozone tendencies. My understanding is that MERRA-2 assimilates both column ozone and ozone profiles. For column ozone, it might make sense to use vertically integrated tendencies, but it does not make sense when ozone profiles are assimilated. Why integrate vertically, if the information is vertically resolved? You would lose information about the altitude profiles. Also, later on

(e.g., figures 6 and 8 and discussion) you yourself use the terms given in equation (1) for specific pressure levels, not vertically integrated.

**Reply:**

The term vertically integrated refers to the nabla operator in the first term of the right-hand side. The continuity equation includes a z-derivative that is usually vertically integrated, which means numerically estimated from the layer above and below. In so far vertically refers only to the 3DVAR data assimilation, which couples the layers above and below and is not meant as vertical integrated quantity for ozone (dobsen units).

**Comment:**
Line 129, equation (1): I did not find this equation in either of the references you provide, also not in Wargan et al, 2015. Please clarify where this equation is from, clearly providing the reference and equation number within this reference.

**Reply:**

The equation was presented in Lubis et al 2017 (page 2439, first paragraph left column (top)) and this paper refers to Bosilovich et al., 2015. Both references are included.

**Comment:**
Figures 3 and 4: why show GROMOS and MLS up to 0.01 hPa, but not MERRA-2? Presumably, the reason is that the agreement is not good. However, you can't evade a discussion of this point by just not showing this altitude region for MERRA-2. I see two options. Either exclude the region 0.1-0.01 hPa completely. Than you shouldn't show it here or in any other figure for any of the data-sets, and make a clear statement in the methods description why this region is excluded from the analysis. Or you can show the region here; than you should include results from MERRA-2 here and in the following for this region as well, and discuss here why they don't agree well with MLS (despite MLS being assimilated into MERRA-2 up to 0.02 hPa) and GROMOS.

**Reply:**

Figure 3 and 4 show all data sets up to an altitude of 0.01 hPa. Only Figure 2 is limited to 0.1 hPa.

**Minor Comment:**
Lines 10-11: it is not clear to me in this context what this means. Unfortunately this also still does not become clear in the description of methods in Section 2.3/2.4. I think what you mean is "..we contrast results from the continuity equation using MERRA-2 reanalysis data with the terms of the ozone tendencies as used in the MERRA-2 ozone assimilation model."?

**Reply:**

We rephrased this sentence and replaced "contrast" by "compare".

**Comment:**
Line 50-53: It is unclear what you are trying to say in this sentence. In particular, "In the mesosphere/lower thermosphere (MLT) region, … in the mesosphere …and the altitude region" is unclear - do you mean "in the MLT" or "in the mesosphere"? Which altitude region?

**Reply:**

We rephrased this sentence and made the altitude region less ambiguous.

**Comment:**

Lines 109-112: "are used to estimate ozone in MERRA-2" they are assimilated into MERRA-2. Please clarify.

**Reply:**

The manuscript uses assimilated:

"The retrieved ozone profiles from the Solar Backscatter Ultraviolet Radiometer (SBUV, 1980 to 2004) and the MLS (since August 2004, down to 177 hPa until 2015, down to 215 hPa after 2015 and up to 0.02 hPa) and TCO from SBUV (1980 to 2004) and the Ozone Monitoring Instrument (OMI) (since 2004) are assimilated into MERRA-2 (Gelaro et al., 2017)."

**Comment:**

Lines 204-207: "our analysis reveals a qualitative agreement …" this is maybe mostly a matter of style, but as this is a conclusion from the results shown in this section, this statement should be at the end of the section, not at the beginning.

**Reply:**

We did draw this conclusion here in this section to justify the more detailed analysis that followed in section 5. As this was a recommendation of another reviewer, we would like to keep it.